



# The secret life of garnets: A comprehensive, standardized dataset of garnet geochemical analyses integrating localities and petrogenesis.

Kristen Chiama[1,2], Morgan Gabor[1], Isabella Lupini[1,3], Randolph Rutledge[1], Julia Ann Nord[1], Shuang Zhang[4,5], Asmaa Boujibar[4], Emma S. Bullock[4], Michael J. Walter[4], Kerstin Lehnert[6], Frank Spear[7], Shaunna M. Morrison[4], Robert M. Hazen[4]

[1] Atmospheric, Oceanic and Earth Science, George Mason University, Fairfax, VA 22030, United States of America.
[2] Department of Earth and Planetary Sciences, Harvard University, 20 Oxford St, Cambridge, MA 02138, United States of America.
[3] Department of Geology, Kansas State University, Manhattan, KS 66506, United States of America.
[4] Earth and Planets Laboratory, Carnegie Institution for Science, Washington, DC 20015, United States of America.
[5] Department of Oceanography, Texas A&M University, College Station, TX 77843, United States of America.
[6] Lamont-Doherty Earth Observatory, Columbia University, New York, N.Y. 10027, United States of America.
[7] Department of Earth and Environmental Sciences, Rensselaer Polytechnic Institute, Troy NY, 12180, United States of America.

*Correspondence:* kchiama@g.harvard.edu (K. Chiama).

**Abstract.** Integrating mineralogy with data science is critical to modernizing Earth materials research and its applications to geosciences. Data were compiled on 95,588 garnet sample analyses from a variety of sources, ranging from large repositories (EarthChem, RRUFF, MetPetDB) to individual peer-reviewed literature. An important feature is the inclusion of mineralogical "dark data" from papers published prior to 1990. Garnets are commonly used as indicators of formation environments, which directly correlate with their geochemical properties; thus, they are an ideal subject for the creation of an extensive data resource that incorporates composition, locality information, paragenetic mode, age, temperature, pressure, and geochemistry. For the data extracted from existing databases, we increased the resolution of several key aspects, including petrogenetic and paragenetic attributes, which we extended from generic material type (e.g., igneous, metamorphic) to more specific rock type names (e.g., diorite, eclogite, skarn) and locality information, increasing specificity by examining the continent, country, area, geological context, longitude, and latitude. Likewise, we implemented a broad silica confidence interval to exclude samples of questionable composition from further analysis. This comprehensive dataset of garnet information is an open-access resource available in the Evolutionary System of Mineralogy Database (ESMD) for future mineralogical studies, paving the way for characterizing correlations between chemical composition and paragenesis through natural kinds clustering. We encourage scientists to contribute their own unpublished and unarchived analyses to the growing data repositories of mineralogical information that are increasingly valuable for advancing scientific discovery.

## 1 Introduction

As scientific discovery becomes increasingly dependent on the internet, older publications are disappearing from the scientific record. Mineral analyses published prior to 1990 are recorded in documents (hard copy journals, books, scanned PDFs, and photographs) that are difficult to convert to a digital format. Without efforts to collect and preserve these data, their value will be lost to the scientific community and become "dark data", information that is not currently accessible in existing geochemical databases or is not represented in the supplementary data of peer-reviewed literature (Hazen et al., 2019; Prabhu et al., 2020). This project emphasizes accumulating dark data with large datasets which both prevents the loss of scientific material and expands the availability of mineralogical data (Hazen, 2014; Hazen et al., 2019; Wilkinson et al., 2016).



The aim of this project is to compile a dataset of geochemical, temporal, and spatial properties pertaining to the garnet mineral
group as a means for data-driven discovery in mineralogy and petrology. Gathering data from existing literature and presenting
the results in an easily accessible manner with tabulated numeric and categorical data provides opportunities for inductive
inference (Hazen et al., 2019; Wilkinson et al., 2016) and abductive discovery (Hazen, 2014). Dark data were collected and
tabulated along with information from established geochemical databases and recent publications to create a comprehensive and
standardized dataset (Chassé et al., 2018; Deer et al., 1982; Gatewood et al., 2015; Hazen et al., 2019; Jochum et al., 2007;
Lehnert et al., 2000; Locock, 2008; Spear et al., 2009; Wilkinson et al., 2016). The resultant garnet dataset consists of 95,588
sample analyses from peer-reviewed literature published between 1949 and 2019. The dataset incorporates 171 diverse attributes
pertaining to locality information, petrogenetic and paragenetic mode, major element oxides, trace elements, isotopic ratios, and
rare earth elements (REEs) as well as additional information when available, such as zonation, color, age, temperature, and
pressure. The creation of this dataset required a series of definitions and assumptions to maximize the amount of information
recorded for each sample without losing the standardization. Specific information regarding each attribute can be found in the
Methods section (Sect. 2). This newly compiled dataset offers the opportunity for researchers to explore the spatial and temporal
history of garnet formation and related geologic processes by using multiple statistical and machine learning techniques,
specifically in the evolutionary system of mineralogy and natural kind clustering (Hazen et al., 2019; Morrison et al., 2020).

## 1.1 Data Integration

Integrating mineralogy with data science is an important step to modernize the field of Earth science. Mineral informatics relies
on robust and cohesive mineral databases (Hazen et al., 2019; Lafuente et al., 2015; Lehnert et al., 2000; Morrison et al., 2020;
Prabhu et al., 2020; Prabu et al. 2022; Spear et al., 2009). Typical examples of existing open-access databases in the
mineralogical community include Mindat, EarthChem, MetPetDB, PetDB, the RRUFF Project, the Mineral Evolution Database
(MED), GeoRoc, and GeoReM (Mindat.org: https://www.mindat.org; EarthChem Portal: http://www.earthchem.org/portal;
MetPetDB: http://metpetdb.com/; PetDB: http://www.earthchem.org/petdb; The RRUFF Project: https://rruff.info/; MED:
https://rruff.info/evolution/; GeoRoc: http://georoc.mpch-mainz.gwdg.de/georoc/Start.asp; GeoReM: http://georem.mpch-
mainz.gwdg.de/; Golden 2019; Jochum et al., 2007; Lafuente et al., 2015; Lehnert et al., 2000; Spear et al., 2009). As
instrumentation improves, high-resolution spatial geochemical data are being continuously produced and additional efforts are
often needed to integrate these new data into the existing databases. Moreover, robust metadata relating to geochemical analyses,
such as temporal and spatial information, are not recorded in the same format across publications and studies, but those metadata
will increase the value of and return on data science in future research. Further, introducing unambiguous location data, such as
detailed categorical locality information combined with specific longitude and latitude coordinates, will increase reliability and
standardization. Therefore, a standardized approach to storing data will solve reproducibility issues that stem from a lack of
documentation and improper representation. Metadata standards in reporting location and spatial data were adopted from
EarthChem as they allow for the seamless integration of metadata from PetDB, GeoRoc, MetPetDB, GeoReM (Lehnert et al.,
2000). Further, there are several efforts underway to produce data standards across the various geochemical and Earth science
data types, including IUGS/CGI (https://cgi-iugs.org/), OneGeochemistry (Lehnert et al. 2019), OneGeology (Jackson 2008), and
OneStratigraphy (Wang et al. 2021).

Due to limited digital documentation, older publications and data are disappearing from the scientific record to become "dark
data." According to Hazen et al. (2019), dark data in mineralogy consists of "information on mineral compositions, localities,
and other data that are available only through hard-copy publications, proprietary corporate documents (notably companies in the
natural resources industry), or privately held research records." For example, garnet sample analyses published prior to 1990 are

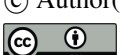



recorded in scanned PDFs that are difficult to convert to an Excel spreadsheet by automated means. These sources of data are not easy to manipulate and often disappear from scientific records with time. Thus, a primary purpose of this study is to record dark

data in a standardized format that is readily accessible, which prevents both the loss of scientific material and continues to expand the availability of mineralogical data.

Standardization of data within the mineralogical community needs to be firmly established. For example, color characteristic names vary dramatically among projects and are subject to the authors' interpretations. Deer et al. (1982) featured descriptive, yet ambiguous, color labels for samples such as "parrot green" which is difficult to integrate into a dataset. In some applications,

specialized systems of color classification have been proposed. For example, the Gemological Institute of America (GIA) has developed a set of standards with descriptive language as well as virtual codes for characterizing specific gem colors (http://gemologyproject.com/wiki/index.php?title=Color_grading, accessed 10 October 2020; Web Colors: https://en.wikipedia.org/wiki/Web_colors, accessed 16 October 2020). In regard to geochemical research, using categorized descriptive terms would allow scientists to convey their data in a more precise and accurate manner. Implementing

standardization practices also enables data from disparate sources to be easily accessed for future evaluation or comparison with other databases.

The Findable, Accessible, Interoperable, and Reusable (FAIR) initiative, while new within the geological community, has been instrumental in bolstering data preservation throughout the physical sciences (Wilkinson et al., 2016). The FAIR Principles for database curation encourage proper data management as well as stewardship across a broad range of disciplines to benefit the

entire academic community (Wilkinson et al., 2016; FAIR Principles: https://www.force11.org/fairprinciples; accessed 14 October 2020). Currently, EarthChem and MetPetDB are advancing data science in geosciences by providing an open-access repository with rich datasets (Lehnert et al., 2000; Spear et al., 2009).

## 1.2 Garnets

Garnets were selected for this dataset owing to their vast informative properties, such as geochemical characteristics, physical

attributes, wide range of paragenetic modes, distribution throughout geological time, resistance to weathering, and resilience during diagenetic processes (Alizai, Clift, and Still, 2016; Chen et al., 2015; Čopjaková, Sulovský, and Paterson, 2005; Deer et al., 1982; Hazen et al., 2008; Kotková and Harley, 2010; Morton, Hallsworth, and Chalton, 2004; Yang et al., 2013). This section will summarize some relevant information pertaining to garnets and their applicability for a comprehensive dataset incorporating localities, petrogenesis and paragenesis, as well as geochemical data.

Garnets are good indicators of formational environments as they contain distinct age, temperature, and pressure information indicative of the protolith chemistry as well as mineral evolution throughout geological time (Baxter, Caddick, and Dragovic, 2017; Baxter and Scherer, 2013; Chen et al., 2015; Deer et al., 1982; Hazen et al., 2008; Kotková and Harley, 2010). For instance, the high-pressure garnet majorite ($Mg_3[MgSi]Si_3O_{12}$) was formed during the era of planetary accretion (>4.56-4.55 Ga) through impact transformations of pyroxene and, subsequently, through igneous and metamorphic processes in Earth's mantle.

Grossular ($Ca_3Al_2Si_3O_{12}$) and andradite ($Ca_3Fe_2Si_3O_{12}$) emerged from the secondary thermal alteration of chondrites and achondrites, potentially very early in solar system history (~4.56 to 4.55 Ga; Fagan et al., 2005; Hazen et al., 2008). There are also reported rare instances of goldmanite ($Ca_3[V,Al,Fe,Ti]_2Si_3O_{12}$), eringaite ($Ca_3Sc_2Si_3O_{12}$), and rubinite ($Ca_3Ti_2Si_3O_{12}$) occurring in chondrite meteorites (Hazen et al., 2008; Morrison and Hazen, 2020). Both grossular and andradite are characteristic of carbonate-bearing metamorphic material; however, formation of andradite depends on the availability of $Al^{3+}$ and $Fe^{3+}$ during

metamorphism (Nesse, 2013). Earth's differentiation, volcanic activity, and plate tectonics gave rise to new garnet species (Hazen et al., 2008). Pyrope ($Mg_3Al_2Si_3O_{12}$) potentially formed through early volcanic processes on Earth's surface from 4.55





Ga to 4.0 Ga (Hazen et al., 2008). Further, pyrope is formed in magnesium-rich, high-grade metamorphic and ultramafic igneous environments and is also commonly found in eclogite and serpentinite (Deer et al., 1982; Nesse, 2013). Almandine ($Fe_3Al_2Si_3O_{12}$) possibly first formed around 4.0 to 3.5 Ga as it is indicative of felsic igneous environments, occurs in medium- to

low-grade metamorphic terrains and is typically found in pegmatites, granite, mica schist, or gneiss (Deer et al., 1982; Nesse, 2013). The initiation of plate tectonics occurred prior to 3.0 Ga followed by the appearance of spessartine ($Mn_3Al_2Si_3O_{12}$) in uplifted regional metamorphic environments (Hazen et al., 2008). Spessartine and almandine-spessartine varieties are common in felsic igneous rocks such as granite and pegmatites in addition to manganese-rich metamorphic rocks (Deer et al., 1982; Makrygina and Suvorova, 2011; Nesse, 2013). Uvarovite is rare and often occurs in chromite-rich igneous environments such as

peridotite or serpentinite (Deer et al., 1982; Nesse, 2013). The complex story of garnet mineral evolution and diverse formational environments provides an excellent case study to investigate the relationship between paragenetic modes, geochemical data, and location information through natural kind clusters (Boujibar et al., 2020; Hazen, 2019; Hazen and Morrison, 2020, 2021; Hazen, Morrison, and Prabu, 2020; Hazen et al., 2008; Morrison and Hazen, 2020, 2021; Nesse, 2013).

In addition to a diverse story of mineral evolution, garnets are often used as geochronometers, geothermometers and

geobarometers (Baxter, Caddick, and Dragovic, 2017). Similar to zircons, garnets are effective in establishing the chronology of geological events by using radiogenic parent-daughter isotopic ratios, such as Sm-Nd, U-Pb, and Rb-Sr (Baxter and Scherer, 2013; Kotková and Harley, 2010). Garnet phase equilibria and mineral-mineral element exchange reactions also provide thermometric and thermobarometric information for a wide range of rock types including during regional metamorphism in crustal protoliths (Baxter and Scherer, 2013; Chen et al., 2015) and in mafic and ultramafic mantle rocks (Nickel and Green,

1985; Nimis and Grutter, 2010; Wu and Zhao, 2011). The majorite content of garnet inclusions provide the only reliable information of the depth of formation in sublithospheric diamonds (Thomson et al., 2021). Garnets often undergo crystal rotation, complex zonation, and deformation, which can be used to distinguish specific grain kinematic histories and shearing planes in metamorphic rocks (Rosenfeld, 1970; Spear and Daniel, 2001; Whitney and Seaton, 2010).

In nature, garnets close to ideal end-member compositions are rare. Therefore, natural samples are often expressed as

percentages of several idealized end-members calculated from the major oxides or oxygen cation ratios (Deer et al., 1982; Geiger, 2016; Grew et al., 2013; Nesse, 2013). According to the International Mineralogical Association (IMA) Commission on New Minerals, Nomenclature and Classification (CNMNC) list of approved mineral species (https://rruff.info/ima/; accessed 5 October 2020), the garnet supergroup contains 37 structural garnet species, while the silicate garnet group consists of six major end-member species and 14 minor species classified by their idealized chemical formula:$X_3Y_2Si_3O_{12}$ (Deer et al., 1982; Grew et

al., 2013). The two main garnet series are pyralspite and ugrandite, both of which form continuous solid-solution series (Deer et al., 1982; Nesse, 2013). Pyralspite consists of pyrope, almandine, and spessartine which requires aluminum in the Y-site while ugrandite includes uvarovite, grossular, and andradite which requires calcium in the X-site (Deer et al., 1982; Nesse, 2013). Historically, it was thought that a miscibility gap exists between the pyralspite and ugrandite series; however, it is now known that uncommon intermediate compositions between the two series exist (Deer et al., 1982; Geiger, 2016; Nesse, 2013).

Additionally, there is some contention about whether these series should be used as they exclude high-pressure garnet species, such as majorite, which are prevalent in the transition zone of the mantle (Geiger, 2016).

The detailed garnet solid-solution series from major oxides ($SiO_2$, $TiO_2$, $MgO$, $MnO$, $FeO$, $Fe_2O_3$, $Al_2O_3$, $CaO$, $Cr_2O_3$, $NiO$, $K_2O$) are classified based on several rules regarding chemical composition. However, the goal of understanding the evolutionary system of garnet group minerals requires a paragenetic context for mineral classification - one that is based on each specimen's

formational conditions, as well as its composition. Recognizing distinct types of garnets thus requires natural kind clustering, which relies on the complex, multivariate correlations among all of the major, minor, and trace element constituents of garnet





samples to determine their paragenetic relationships (Hazen et al., 2019; Morrison and Hazen, 2020). To that end, we initiated this study to establish an extensive, reliable, open-access data resource of garnet sample analyses across a multitude of resources for data pertaining to geochemistry, localities, and petrogenetic and paragenetic modes. This paper does not attempt to define the

compiled garnet sample analyses by the IMA classification of end-member compositions, but rather aims to preserve the raw data in order to prevent potential bias in future analyses.

## 2 Methods

We compiled a dataset of 95,588 garnet analyses across a total of 171 attributes (doi: 10.48484/camh-xy98). The dataset includes 61,294 analyses from EarthChem (doi: 10.26022/IEDA/112171; 64 from NAVDAT, 47,591 from GeoRoc, and 13,639 from

PetDB), 12,781 from Chassé et al. (2018), 10,380 almandine point analyses from the supplementary data in Gatewood et al. (2015), 6,787 samples from MetPetDB (doi: 10.26022/IEDA/112173), 4,070 assorted samples from peer-reviewed literature and other datasets such as the RRUFF project, and finally 275 original electron microprobe analyses (EMPA). Peer-reviewed literature was compiled in Zotero (https://www.zotero.org/; accessed 14 October 2020) and sample analyses were converted from PDF documents to Excel using Tabula (https://tabula.technology/; accessed 27 September 2020) or by manual entry, depending

on the quality of the PDF. This section will examine the methods and assumptions behind the formation of the dataset as well as the methods employed to analyze 9 original garnet samples.

### 2.1 Dataset Formation

The primary attributes incorporated in the dataset include locality information, petrogenesis and paragenesis, as well as major oxides. Secondary attributes include the sample age, temperature, pressure, trace elements (e.g., REEs), and isotopes when

provided by the source material. Each of the attributes are identified in a detailed system while maintaining the ability to cluster and identify patterns within the dataset. A data schema is included in Table 1 to define each of the attributes in order of appearance in the dataset.

| Table 1. Description of Attributes Present in the Dataset | | | | |
|---|---|---|---|---|
| **Attribute Name** | **Full Name** | **Definition** | **Datatype** | **Attribute Dependent Groups** |
| Project ID | Project ID | Sample analysis line number. | Integer | Sample Identification |
| IGSN | International GeoSample Numbers (IGSN) | International GeoSample Numbers (IGSN) for each of the original EMPA garnet analyses. | String | |
| Indiv. Project ID | Individual Project ID | Line number paired with an indicator of where sample information originated from such as the major data repositories' or the initials of the author who compiled the samples from peer-reviewed literature. EC_GARNET = EarthChem; MetPetDB; Chasse et al., (2018); Gatewood et al., (2015). | String | |
| Origin ID | Original ID | Original ID labels based on their respective data repository or literature sources. | String | |
| Repeat | Repeated Sample Information | A '0' and '1' flag for repeated sample information between data sources. A '0' is the first iteration of sample information and '1' is the second iteration of sample information. | Integer | |



| Mineral | Mineral Name | Dominant silicate garnet group species, structural garnet group species, garnet end-member species or end-member combination name. 39 total species name variations. Unidentified samples listed as 'Garnet' for clarity. | Categorical | |
|---|---|---|---|---|
| Varietal Name | Mineral Species Varietal Name | Any additional garnet species or varietal species information. | Categorical | |
| Hydrated Garnet | Hydrated Garnet | A '0' and '1' flag for whether samples were identified as hydrated in the original literature. '0' indicates non-hydrated and '1' is hydrated. | Integer | |
| Zone | Zonation | Indicates the concentric zone sample analyses were taken from in a grain. Simplified to the core (c), middle (m), and rim (r) of each grain. | Categorical | |
| Location | Detailed Location | Detailed location taken verbatim from the sources. | Categorical | |
| Continent | Continent | The continent from which each sample was collected. | Categorical | |
| Country | Country | The original country name (at the time of collection). | Categorical | Location Information |
| Area | Area | Records more specific locality information encompassing regions, provinces, states, districts, and counties. | Categorical | |
| Geological Context | Geological Context | Records more specific information concerning the geological formation environment of the collection site such as metamorphic terranes. | Categorical | |
| Latitude | Latitude | Measured in decimal degrees. | Integer | |
| Longitude | Longitude | Measured in decimal degrees. | Integer | |
| Title | Title | Title of the paper that sample analyses originated from. | Categorical | |
| Journal | Journal | Journal the paper was published in. | Categorical | References |
| Reference | Reference | Authors of the paper sample analyses were published in and year of publication. Original References formatting from EarthChem and MetPetDB was maintained. | Categorical | |
| Formation | Formation environment (geological) | Detailed formation environment obtained verbatim from the sources. | Categorical | |
| Material | Material | Denotes whether the parent material of each sample is classified as Detrital, Igneous, Metamorphic, Extraterrestrial, Metasomatic, or Unknown. | Categorical | |
| Type | Type | Details the type of material from which samples originated. For example, the type of igneous material is identified to be Volcanic, Plutonic, etc., whereas the type of metamorphic material examines metamorphic facies such as Amphibolite, Greenschist, Eclogite, etc. | Categorical | Petrogenesis |
| Composition | Composition | Dominant mineral assemblages, such as Felsic, Mafic, Ultramafic, Carbonate, or Calc-Silicate etc. | Categorical | |
| Paragenesis | Paragenesis | Specific rock-type name; a one- or two-word term that adequately represents the sample. Rock-type definitions and classifications were taken verbatim from the literature as well as Mindat as it is a well-accepted database in mineralogy for classification. | Categorical | |
| Analysis Method | Analysis Method | Instrumentation used for chemical analysis, often EMPA or LA-ICP-MS. | Categorical | |
| GIA Hue | Gemological Institute of | Hue or shade of the sample. | Categorical | Color |



| | | | | |
|---|---|---|---|---|
| | America Hue | | | |
| GIA Tone | Gemological Institute of America Tone | Level of grayscale within the color. | Categorical | |
| GIA Saturation | Gemological Institute of America Saturation | Intensity of the color. | Categorical | |
| Min Age (Ma) Youngest | Minimum Age in Ma | Minimum Age in Ma. | Integer | Age |
| Sample age (Ma) | Average Age in Ma | Average Age in Ma. | Integer | |
| Max Age (Ma) Oldest | Maximum Age in Ma | Maximum Age in Ma. | Integer | |
| Min P (kbar) | Minimum Pressure in kbar | Minimum Pressure in kbar. | Integer | Pressure |
| P (kbar) | Average Pressure in kbar | Average Pressure in kbar. | Integer | |
| Max P (kbar) | Maximum Pressure in kbar | Maximum Pressure in kbar. | Integer | |
| Min T (°C) | Minimum Temperature in °C | Minimum Temperature in °C. | Integer | Temperature |
| T (°C) | Average Temperature in °C | Average Temperature in °C. | Integer | |
| Max T (°C) | Maximum Temperature in °C | Maximum Temperature in °C. | Integer | |
| Notes | Notes | Notes are individual per sample. The presence of birefringence, inclusions, twinning, crystal shape, original references, and original categorical color designations are included for the respective sample when provided. | Categorical | |
| Confidence Interval of $SiO_2$ (wt%) | Silica Confidence Interval in weight percent | Distribution of silica content in the dataset are classified into a quality control (A, B, C) to exclude problematic samples from further analysis. There are 73,868 'A' quality garnet samples within 2 standard deviations from the mean, followed by 639 'B' samples within 3 standard deviations, and finally 1,511 'C' samples outside 3 standard deviations. It is recommended to maintain A and B samples based on the natural diversity of garnet species but exclude or check the influence of C samples. The ranges of $SiO_2$ weight percent (wt%) are as follows: A: 33.059 - 47.746 wt% B: 29.387 - 33.059 wt% and 47.746 - 51.418 wt% C: $\leq$ 29.387 wt% and $\geq$ 51.418 wt% | Categorical | |
| Our Calc (wt%) | Our Calculation of the Sum of Major Oxide Totals in weight percent | Sum of all recorded major oxides for each sample, excluding ones that listed oxides in two forms (ex. if FeO and FeOT were both listed only one was used in the calculation). | Integer | |

**Table 1. Descriptions for each of the attributes in the dataset by order of appearance.**

Data were compiled from multiple resources to create this dataset. The data were extracted from the EarthChem Portal database

which provides a central access point to mineral composition data from PetDB, GeoRoc, and NAVDAT by querying for all garnet analyses available ('analyzed material' = 'garnet') and retrieving all available variables (date downloaded: 13 Aug. 2019). Data from MetPetDB were compiled from a search for chemical analyses of garnet and a search for samples that contain garnet. The two searches were then cross correlated by the original sample ID so that each garnet analysis could be annotated with



location, rock type, and other metadata (date downloaded: 24 Dec. 2020). Majorite samples are from the compilation of Walter et al. (in press). All other samples were compiled by undertaking a literature review of garnet sample analyses which provided geochemical data, geologic formation environment, and/or location information. The data from the data repositories and literature were standardized for common attributes to form the structure of this dataset.

We created an identification system to maintain as much information as possible from original sources and additional references. Each sample was given a unique 'Project ID' which is indicated by a line number to identify the total number of samples examined. The 'Individual Project ID' indicates where the major data repositories' sample information originated from (i.e., EarthChem employs a line number followed by EC_GARNET) or the initials of the author who compiled the samples from peer-reviewed literature. Multiple sources did not provide International GeoSample Numbers (IGSN; https://www.igsn.org/; accessed 27 September 2020), however, the original EMPA garnet sample analyses performed in this study were assigned IGSNs. The 'Origin ID' attribute was created to label sample analyses based on their respective original sample identification.

A detailed reference section was embedded in the dataset for future researchers to quickly locate the original source of samples. This section was split into three separate attributes: Title, Journal, and Reference. The 'Reference' attribute lists the authors and year of publication while maintaining the formatting for the samples originating from the EarthChem and MetPetDB repositories. The 'Title' and 'Journal' attributes were adopted to prevent confusion because some authors published multiple papers on garnet samples in the same year; for example, Chassé et al. (2018) reported samples from Griffin et al. (1999A and 1999B). This multi-attribute referencing and identification system was adopted to quickly identify any additional information regarding specific samples not already included in the dataset. Reference formats from EarthChem and MetPetDB were maintained to simplify cross-referencing.

### 2.1.1 Mineral Species

Regarding the IMA classification of garnet species, there are 37 minerals within the garnet structural group, 14 garnets within the silicate group, and 6 common end-member species (https://rruff.info/ima/; accessed 5 October 2020). As it is not within the scope of this paper to apply the IMA classification of composition for each sample, we simply assigned a dominant garnet species name if one was reported. Often, many literature sources and data repositories (EarthChem and MetPetDB) will not classify a garnet sample by a specific species as garnets are typically chemically zoned. We indicated all unidentified samples as 'Garnet' which dominates the dataset (82,558 analyses). Samples reported as a combination of end-members were listed as both (i.e., 'Almandine-Spessartine'; Yang et al., 2013). There are a total of 39 possible variations of mineral species in the database (including the unknown 'Garnet' flag) defined by 6 end-members, 6 silicate group garnets, 21 different combinations of end-members, 4 structural garnet species (bitikleite, elbrusite, henriermierite, and toturite), as well as a chromite inclusion found in the uvarovite sample of the original EMPA analyses which was labeled as such to separate these analyses from further consideration. When an additional varietal species or minor species was provided in the literature, it was recorded in the 'Varietal Name' attribute (i.e., 'Chromian Andradite,' or 'Titanian Melanite'; Deer et al., 1982; Ghosh and Morishita, 2011). Further, hydrated garnets were denoted with a '1' while unhydrated garnets are represented with '0' in the 'Hydrated Garnet' attribute. It is important to note that we recorded samples as hydrous only when samples were denoted as such in the literature.

### 2.1.2 Zonation

Garnets are often highly chemically zoned throughout each grain, and the zonation can be used to understand the changing environmental conditions, such as temperature and pressure, over time (Javanmard et al., 2018; Yang et al., 2013). Although there is debate about the complexity and style of zonation within garnet samples, it is not within the scope of this paper to





address zonation in detail. This section will address different types of zonation leading to a discussion about how to use the 'Zone' attribute in the dataset.

Classically, zonation for garnets is measured concentrically from the core to rim of the grain (Javanmard et al., 2018; Yang et al.,
2013). Polycrystalline garnets, though less common, can record the changing mechanisms and chemical conditions by combining 2 to 30+ crystallites within one garnet grain (Whitney and Seaton, 2010). The major divalent cations in garnets (Fe, Mg, Mn, and Ca) can feature different styles of zonation within individual polycrystals (Spear and Daniel, 2001; Whitney and Seaton, 2010). This style of zonation leads to classification issues in a dataset format, such as identifying specific styles of zonation across multiple studies and classifying them with limited information. For example, polycrystalline zonation is identified by polycrystal
number while concentric zonation is classically identified by zone number originating from the core and increasing in numerical value towards the rim (Whitney and Seaton, 2010).

We intended to maintain as much information as possible about the individual samples without over-complicating the dataset through the zonation classification process. Yet, many authors and databases did not report zonation or only reported core, middle, and rim of each grain and did not interpret polycrystalline zonation. Therefore, while zonation is crucial to identifying
the mechanisms and paragenetic conditions of garnet formation, we cannot identify polycrystalline or complex zonation from limited data. Ultimately, the 'Zone' of each sample analysis was classified simply by the core (c), middle (m), and rim (r) of each grain. For samples that were unclear or did not report zonation, this field was intentionally left blank. Ideally, a standardized system of zonation representation should be adopted to limit the subjectivity and interpretation of zones. The clarity would have allowed us to adopt a dual-attribute system identifying the style of zonation (e.g., concentric, polycrystalline) in one attribute for
each point analysis and the polycrystal or concentric zone number in a second attribute. This system would proffer a more in-depth analysis of compositional evolution across complex zonation styles.

### 2.1.3 Locality

Locality information from the literature and repositories varies dramatically in specificity. In order to maintain continuity, the location information was classified into four categories: Continent, Country, Area, and Geological Context. In the cases where a
country or regional area has politically dissolved, the original published nomenclature for each sample was maintained in either the 'Location' or 'Country' attribute to prevent confusion over historical borders. For example, Deer et al. (1982) references former countries such as the USSR and Czechoslovakia. The 5 extraterrestrial samples are recorded by the location they were discovered (Continent, Country, and Area) and are designated as extraterrestrial material in the petrogenetic attributes. The regional 'Area' encompasses provinces, states, districts, counties and cities while the attribute 'Geological Context' focuses more
specifically on the geological location information such as metamorphic terranes, kimberlite fields, and mining sites. Some sources provided a further in-depth description or information that did not fit into these designated categories (Deer et al., 1982; Herbosch et al., 2016). To prevent oversimplification, any additional information was denoted in the 'Location' attribute. Latitude and longitude were converted from degrees, minutes, and seconds to decimal degrees for ease of use.

### 2.1.4 Petrogenetic Attributes

The categorization of geological and mineralogical formation environments was a key component in the formation of this dataset. We define petrogenesis as the origin and formational conditions of the host rock and paragenesis as a characteristic rock-type name associated with the origin and formation conditions of minerals based on definitions obtained from Mindat.org (https://www.mindat.org/; accessed 30 December 2020). Because petrogenesis and paragenesis are reported differently between studies, a standardized system was required to adequately categorize this information in a dataset format. The goal of the



petrogenetic attribute classification system was to organize data for resolution-dependent cluster analysis. All of the sample analyses were identified by a series of petrogenetic attributes such as: a detailed geologic 'Formation' environment, general parent 'Material', 'Type' and 'Composition' of parent material, and finally a general 'Paragenesis.' These attributes were chosen such that petrogenetic and paragenetic clusters can be examined with different degrees of resolution.

The detailed Formation environment is different for nearly every sample as it was extracted verbatim from the peer-reviewed literature; thus, this attribute has the highest resolution. In contrast, the Material attribute offers the lowest resolution as it was simplified to detrital, igneous, metamorphic, extraterrestrial, metasomatic, and unknown material from which the samples originated. Type describes the type of material from which samples originated. For example, the type of igneous material was identified to be volcanic or plutonic, whereas the type of metamorphic material examined metamorphic facies such as amphibolite, greenschist, and eclogite facies. The Composition focused on the dominant mineral assemblages primarily related to igneous and metasomatic materials, such as felsic, mafic, ultramafic, carbonate, and calc-silicate. Therefore, the Composition attribute was simplified to represent information that can be identified across most peer-reviewed literature. Because not all studies reported specific mineral assemblages, it is not within the scope of this paper to assign and classify the associated minerals by locality. Regarding the Paragenesis attribute, a majority of previous publications classify paragenesis as a detailed mineral formation process which does not translate to a dataset format that can be clustered. Thus, the attribute Paragenesis was simplified to the rock-type name; a one- or two-word term that adequately represents the sample. Rock-type definitions and classifications were taken verbatim from the literature as well as Mindat.org as it is a well-accepted resource for mineralogy (https://www.mindat.org/; accessed 30 December 2020).

This petrogenetic attribute reporting system offers the opportunity for resolution-dependent cluster analysis. Material is the lowest resolution attribute containing only six categories while Paragenesis is the highest resolution attribute representing 174 different paragenetic modes. We recommend examining each of the petrogenetic attributes collectively as well as individually to best characterize the data with cluster analysis.

### 2.1.5 Age, Pressure and Temperature

Samples that reported age (Ma), pressure (kbar), and/or temperature (ºC) of formation were recorded in the dataset, including uncertainty, when provided. Each of these parameters included attribute columns with standardized units for the minimum, average, and maximum value. Despite garnets being excellent environmental indicators, few sources reported a specific formation temperature, pressure, or age for sample analyses. Rather than directly dating the garnet grains, some studies reported the broad ages of metamorphic terranes which can be constrained by various methods (e.g., stratigraphy, igneous ages, accessory mineral ages). These samples and ages were not further examined within the dataset as our goal was to preserve the raw data. Sources that reported detailed age information often reported average values without uncertainty or employed unclear terminology. For example, Herbosch et al. (2016) did not include a degree of uncertainty regarding ages and Parthasarathy et al. (1999) reported ages in terms of epochs or periods which were instead denoted as maximum and minimum dates to maintain consistency in the dataset. Histograms were created for the attributes pertaining to average age, temperature, and pressure which are further discussed in Sect. 3.4.

### 2.1.6 Geochemical Data

A major component of the dataset consists of geochemical information for major oxides and trace elements which account for 129 attributes of the total 171 represented. Major oxides were recorded in weight percent (wt%) whereas trace elements were recorded in parts per million (ppm) to maintain consistency. Generally, older publications reported major oxides to cation



numbers based on 24, 12, or 8 oxygen atoms and/or mole percent end-member species (Deer et al., 1982). We chose to exclude the oxygen cation data and end-member calculations from this dataset as both can be calculated from the major oxides.

Additionally, a few sources provided information on isotopes which were included in the dataset. As some sources did not have a field for the sum of the total oxides, we added an attribute named 'Our Calc (wt%)' which is a summation of all the major oxides to address this issue. This attribute helps identify problematic samples with an abnormally high or low total wt%, which could be misrepresented due to a typographical error, miscalculation, or experimental error.

Additionally, during the acquisition of data, many dark data sources could not be automatically converted to Excel spreadsheets,
therefore, the data were entered manually. Data from Deer et al. (1982) were poorly converted in Tabula (https://tabula.technology/; accessed 27 September 2020) with decimal places replaced by multiplication symbols or values transposed throughout the resulting spreadsheet. Manual entry aimed to prevent data corruption, but this also introduced the opportunity for typographical errors. Data entered manually were double checked for errors using the 'Our Calc (wt%)' column as a summation of the major oxides.

**2.1.7 Iron**

Iron can be found in garnets as $Fe^{2+}$ in the X site of the mineral structure, $Fe^{3+}$ in the Y site, or in both depending on the garnet species (Deer et al., 1982; Nesse, 2013). However, without applying the flank method (Höfer et al. 2000), EMPAs cannot measure the two valences concurrently (Droop, 1987). Instead, most authors assumed all iron to be one chosen valence, resulting in it being recorded as either FeOT (total) when it was all calculated as $Fe^{2+}$, or $Fe_2O_3T$ (total) when all the iron was calculated as
$Fe^{3+}$. Very few studies conducted post-EMPA calculations in order to find both iron oxides for their samples. Additionally, many of the databases presented their iron data in a way that made it unclear if this calculation was performed as they labeled all their analyses as one of the iron oxides yet did not mention the other (Chassé et al., 2018; Gatewood et al., 2015; MetPetDB). As a result, we included four separate columns for iron: 'FeO,' 'FeOT,' '$Fe_2O_3$,' '$Fe_2O_3T$.' However, it was difficult to compare garnets across four attributes for two iron oxides (FeO and $Fe_2O_3$).

In order to evaluate our original EMPA samples, we utilized a spreadsheet created by Locock (pers. comm.), based on the work of Droop (1987), to calculate both FeO and $Fe_2O_3$ from FeOT. The spreadsheet applies the ideal cation:oxygen ratio of garnets (8:12) and the major oxide results (including FeO) to estimate FeO wt%, $Fe_2O_3$ wt%, a new analysis total, and the added amount of oxygen from the presence of $Fe^{3+}$ (which is included in the 'Notes' column of the dataset). This spreadsheet was not applied to the entire dataset for a couple of reasons. First, many of the analyses did not include finite values and reported the concentration
as below the detection limit using '<' or one of several abbreviations for absent or non-detected oxides and trace elements. The spreadsheet cannot interpret these abbreviations; therefore, they had to be removed. One approach to make these data readable by the spreadsheet would be to replace these abbreviations with absolute values, however, this would misrepresent the true values of the data and potentially bias the results. This concept is further described in Sect. 2.1.12. Secondly, the calculation is not suitable for hydrogarnets, which have variable numbers of oxygen atoms per anhydrous formula unit (Droop, 1987). Thus, the
recalculation was only applied to the original EMPA analyses performed in this study.

**2.1.8 Silica Confidence Interval**

According to Deer et al. (1982), the silica content of garnets can range from ~26 wt% to ~58 wt% depending on the mineral species. For pyralspite and ugrandite garnets, $SiO_2$ ranges from ~34 to 44 wt% and is strongly dependent on the amount of (Mg + Al) versus (Fe + Mn + Ca + Ti + Cr). In addition, hydrogarnets have been reported with $SiO_2$ < 27 wt%, while majoritic garnets
can approach 58 wt% $SiO_2$. However, some analyses included in the dataset reported unreasonable $SiO_2$ content, ranging from



0.00 wt% to 99.98 wt%, indicating that some of the reported samples are not garnet and should be excluded from further analysis. Bounding criteria were carefully selected to include a majority of garnet analyses but exclude potential non-garnet inclusions. The 76,018 garnet samples that report $SiO_2$ are identified by two and three standard deviations from the mean to account for the diversity in garnet mineral species outlined by Deer et al. (1982). Of the total samples that report $SiO_2$, 97.17%

are 'A' samples, 0.84% are 'B' samples, and 1.98% are 'C' samples. The samples identified with an A or B are garnets, potentially with different inclusions, formation environments, complex zonation, or diverse varietal species, whereas C samples are likely not garnets. The calculations performed for the $SiO_2$ confidence interval are included with the dataset as well as in Supplement A.

### 2.1.9 Duplicate Samples

Because garnet data were derived from individual studies as well as databases, there was a potential for overlap. Repeated samples were identified by their 'Origin ID,' original references, and identical geochemical information. Only 7.63% of samples are repeated in the overall dataset. The major sources of sample overlap occur with Chassé et al. (2018) and EarthChem. The major difference between these sources is that Chassé et al. (2018) reported categorical location information, whereas EarthChem provided only longitude and latitude. To maintain relevant information, the attribute 'Repeat' was created to list the

first iteration of samples as '0' and the second iteration of samples, or duplicates, as '1' such that samples marked by '1' are excluded from further analysis.

### 2.1.10 Color

Color classification is ambiguous because color definitions are subjective between different authors. Color was the most diverse descriptor of all attributes within our dataset. For example, Deer et al. (1982) reported color in a plethora of different

designations such as "Dark Peach-Tan," or "Hyacinth Red." The method used to standardize the 'Color' column into a clusterable format was adopted from the GIA's (Gemological Institute of America) color grading system, specifically the Gemology Project (http://gemologyproject.com/wiki/index.php?title=Color_grading; accessed 10 October 2020). This system assigns abbreviations to hues and employs numbers to indicate the strength of the tone and saturation for the colors. When saturation or tone were not given as descriptive labels, neutral values were chosen to represent the sample. Typical notation for

the sample is indicated as "hue tone/saturation." For example, "bright green" would be "slyG 5/6." However, for this dataset, each of the three descriptors were separated into individual columns. Because color descriptions are open to interpretation, adapting them to the GIA format without access to the specimens introduces significant room for error. Establishing a universal or standardized color code would be beneficial for conveying exact colors in a non-visual format. We propose a more specific method of characterizing and defining color through virtual color codes, such as Hex, HTML, CMYK color codes, or HSL or

RGB values (https://htmlcolorcodes.com/; accessed 10 October 2020). Virtual color codes are an internationally recognized and accessible format for color grading to limit ambiguity and interpretation error. In our circumstance, we did not have access to the original samples and thus could not identify colors with specific labels.

### 2.1.11 Notes

The 'Notes' column is dedicated to any important sample information that is not regularly reported in established databases or

peer-reviewed literature. For example, the presence of birefringence, inclusions, twinning, crystal shape, and original color designations are noted for the respective sample when provided. Additionally, the original references are recorded in this section if a larger, more encompassing paper or database was the main reference cited. For example, Deer et al. (1982) is a compilation





of sources, so references to the original literature were listed in our 'Notes' column. This approach is also employed by Chassé et al. (2018) and EarthChem, which contain samples compiled across multiple sources and indicate the original authors.

**2.1.12 Analysis Method and Minimum Detection Limit**

Information about instrumentation used in geochemical analyses of garnet samples was recorded in order to avoid interlaboratory biases generated by systematic differences between various equipment (Hazen, 2014). Due to the range in analytical methods, certain terms were used for absent or non-detected oxides and trace elements. The terms found in literature include: below detection limit (bdl, b.d.l.), not detected (nd, n.d., nd., n. d.), not applicable/analyzed (na, n.a.), no value ( - , . , nil), trace (tr, t.r.,

tr.), and '<[VALUE]'. Terms were standardized (e.g., from 'b.d.l.' to 'bdl') to maintain consistency in the dataset. Standardized terms in the dataset include below detection limit (bdl), not detected or not applicable (na), trace (tr), and '<[VALUE]'. Because each one of these abbreviations has a separate definition, we did not significantly alter these terms to prevent misrepresenting the data. For example, 'bdl' could not be replaced with a zero or removed, as it does not explicitly say the oxide or element was not found, simply that it was below the detection limit. Trace values were treated similarly, as standardization of these abbreviations would also not be conducive to representing information from the original sources accurately.

would also not be conducive to representing information from the original sources accurately.
Other concerns included the minimum detection limit for each analysis method. Initially, we examined the minimum detection limit, which ranged in numerical value and varied dramatically among the instrumentation used and the year when various studies were conducted. This information was not included as it could not be standardized nor applied to the entire dataset without altering or potentially skewing the dataset to a particular value.

**2.2 Electron Microprobe Analyses**

In addition to samples compiled in the dataset, major elements from nine garnet samples (almandine, andradite, two samples of grossular, spessartine, uvarovite, and three unknown samples of garnet) donated by George Mason University were measured using a JEOL JXA-8530F Field Emission Electron Microprobe (EMPA) at the Carnegie Institution for Science's Earth and Planets Laboratory in Washington, DC. The microprobe was standardized using albite, $TiO_2$, $MgCr_2O_4$, orthoclase, spessartine-

almandine, pyrope-almandine, and augite. The acceleration voltage was 15kV with a probe current of 20nA and a 5-micron diameter beam. Samples were analyzed for their concentration of Na, Si, Ti, Ca, Mg, Al, Cr, K, Fe, and Mn, and were reported in their oxide form in the dataset. Oxygen was determined by stoichiometry. Each point analysis is identified with an IGSN in the dataset. Additionally, the 'Origin ID' for each analysis was provided to help delineate zonation identified in the samples. Specifically, we identified inclusions within two samples (uvarovite and almandine) that potentially exhibit complex rather than

concentric zonation. The individual sample IDs employ A, B, C to denote the different regions/inclusions measured in these point analyses. However, to maintain consistency with the rest of the dataset, the 'Zone' attribute identifies the location of point analyses in the core, middle, and rim of the grain while inclusion information was classified in the 'Notes' attribute. A total of 275 point analyses were performed with a minimum of 25 points for each sample. In the case of uvarovite which exhibited concentric zonation visible to the naked eye, an additional 24 point analyses were performed in a linear path from the core to the

rim of the grain to confirm the complexity of zonation. A detailed evaluation of the 275 point analyses is included in Supplement B and a summary of the average major oxide concentrations is in Supplement C.

**2.3 Scatterplot Matrices**

Five separate scatterplot matrices were generated with the purpose of observing the relationships between the different elements comprising garnets from five different material types (Metasomatic, Detrital, Metamorphic, Igneous, and Unknown). The





extraterrestrial material type amounted to only 5 samples, and therefore were excluded because of too few analyses to observe any substantial trends. Ten of the major elements that occur within garnets were selected to represent the data in the plots (Si, $Ti^{4+}$, $Al^{3+}$, $Fe^{3+}$, $Fe^{2+}$, Mg, Ca, Na, $Mn^{2+}$, and $Cr^{3+}$). These elements were recorded in weight percent and were recalculated from the major oxides to the individual elements. The raw geochemical data compiled were significantly cleaned in order to accurately conduct a comprehensive and standardized chemical analysis of the dataset. When filtering the garnet analyses, only samples

with a silica confidence interval of A or B were chosen in order to filter out potential non-garnet inclusions. Samples that only reported one oxide analysis or included symbols such as '<' were omitted from further data analysis. Samples expressing information through terms such as 'bdl' or 'tr' were disregarded because R (R Core Team, 2021) cannot interpret these values. The scatterplot matrices from this dataset were produced using RStudio 4.0.5 with three data visualization packages: ggpubr, ggplot2, and GGally (Kassambara, 2020; R Core Team, 2021; Schloerke et al., 2021; Wickham, 2016). The data is

characteristically non-normal in nature, therefore, rather than using the default Pearson correlation coefficient calculation, Kendall's Tau was applied as it is inherently insensitive to errors and outliers.

**3 Results and Discussion**

The analysis of our dataset examines the representation of mineral species, petrogenetic attributes, locality information, and geochemical data of samples while considering the possibility for errors or bias. The purpose is to visualize the compiled data

through single attribute-based diagrams and scatterplot matrices. The mineral species, locality information, and petrogenesis results may be biased due to the sources of compiled data. The standardization and quality of garnet samples were evaluated by the 'SiO₂ Confidence Interval' attribute, which revealed a bimodal distribution due to the compositional differences among garnet species. Regarding the geochemical information presented in the dataset, we investigated the major element compositions of sample analyses with scatterplot matrices.

**3.1 Mineral Species**

This dataset includes the IMA nomenclature to identify the dominant 'Mineral' species for sample analyses. Although it was not our goal to classify garnet species based on composition, the percent of garnet end-members can be calculated through oxygen cation ratios if desired. There are 37 IMA-recognized structural garnet species and 14 silicate garnets, however, there are 39 categories of mineral names within the dataset which includes the combination of end-members such as 'Almandine-Grossular'

and 'Almandine-Pyrope' for samples near 50-50 in composition as well as the simplified term 'Garnet' for unidentified samples. Literature and data sources that reported analyses as a dominant mineral species were recorded as such in the dataset.
The representation of 39 different variations of mineral species in the dataset was plotted by counts of unique categories with two breaks in the scale to prevent the large number of almandine and general garnet samples from obscuring the distribution of the other species present (Fig. 1). Of the 95,588 total sample analyses in the dataset, 82,558 are categorized as general garnet while

13,030 contain more specific silicate and structural garnet species or end-member combination names. The 82,558 unidentified 'Garnet' samples originate from 61,294 EarthChem samples, 12,781 samples from Chassé et al. (2018), 6,787 MetPedDB, and other compiled peer-reviewed literature which did not provide specific garnet species names due to the common chemical zonation of garnets. There are 10,603 samples categorized as almandine, of which 10,380 analyses are from 10 garnet grains described as "dominantly almandine ($X_{Fe} = 0.52$-$0.78$), with subordinate amounts of pyrope ($X_{Mg} = 0.03$-$0.12$), spessartine ($X_{Mn}$

$= 0.00$-$0.25$), and grossular ($X_{Ca} = 0.12$-$0.21$)" by Gatewood et al. (2015). These samples were grouped as general almandine because the primary focus of the dataset was to report raw data, not to further examine the IMA mineral classifications. The



remaining 2,427 sample analyses in the dataset consist of 889 spessartine, 385 andradite, 267 almandine-spessartine, and 886 analyses distributed across 34 other silicate and structural garnets as well as end-member name combinations (Fig. 1). While this distribution is not representative of garnet species in nature, it is significant for the dataset to include as many garnet sample

analyses as possible. It is important to note that the majority of sample analyses are tabulated under the general 'Garnet' flag as the IMA chemical classification scheme can be applied to assign specific percent end-member species to these analyses. Nevertheless, the purpose of this study was not to assign mineral species names but rather to record the raw data provided in the literature and data repositories to provide a comprehensive, standardized database of garnet geochemical analyses.

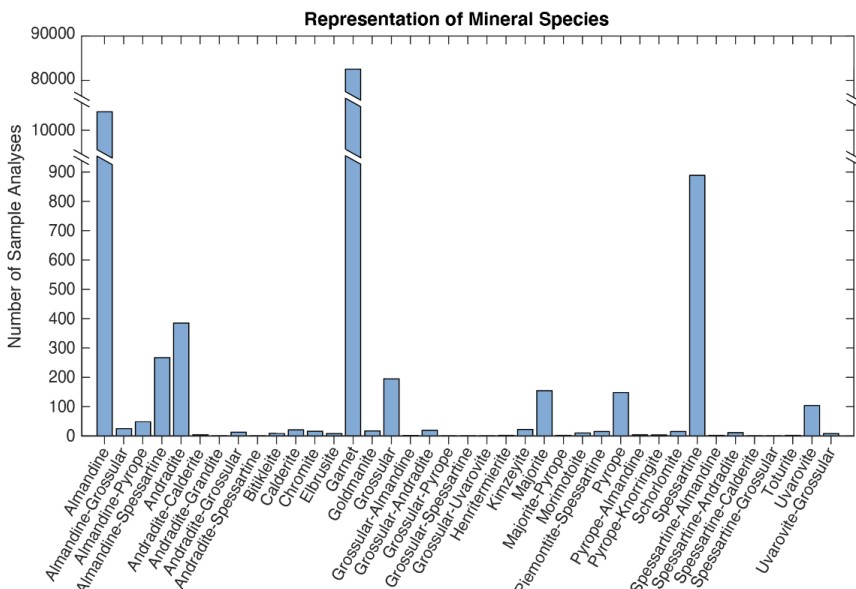

**Figure 1. Representation of all the sample analyses across the 39 different 'Mineral' categories including garnet end-members, end-member combinations, silicate garnets, and structural garnets present in the dataset. There are two breaks in the scale to include 10,603 Almandine and 82,558 general Garnet sample analyses without obscuring the distribution of other categories present. There are 889 spessartine, 385 andradite, and 267 almandine-spessartine analyses as well as 886 analyses accommodated by the remaining 34 categories.**

### 3.2 Locality Information


Locality information within the dataset consists of six attributes of increasing resolution: Continent, Country, Area, Geological Context, Latitude and Longitude. Of the total 95,588 sample analyses in the dataset, up to 33,228 report some form of categorical location information (continent, country, area, or geological context) and 68,364 report numerical data (longitude and latitude), while only 7,972 report both categorical and numerical location data. All sources provided either categorical or numerical

location information except for Locock (2008) which did not contain location data. Thus, a dual system of categorical and numerical location data was created to best represent the entire distribution of sample localities.

There are 33,228 sample analyses that report an origin from one of the seven continents and 32,752 analyses which indicate a specific country of origin. There are 712 unique regional areas represented by 29,013 sample analyses and 397 unique geological contexts for 30,575 sample analyses. The regional area and the geological context attributes include specific locality information

as descriptive as "60 km NW of Kimberley, Cape Province" and "Markt Kimberlite, Subcontinental lithospheric mantle, Rehoboth Subprovince" respectively to increase reproducibility and availability of data (Chassé et al., 2018; Deer et al., 1982). Further, the 5 analyses with an extraterrestrial origin can be identified by the 'Material' attribute and are listed by the continent and country in which they were discovered. The remaining analyses in the dataset, (62,360 continent, 62,836 country, 66,575 area, and 65,013 geological context) did not report location information and are designated as unknown. The distribution of

samples from each continent and country were plotted by counts of unique categories (Fig. 2 and 3). The regional area and geological context attributes were not plotted due to the vast quantity of unique categories. The 68,364 samples that report



latitude and longitude were plotted in R to visualize the global distribution of samples in the dataset which represent 1,786 unique locations (Fig. 4; R Core Team, 2021). Ocean floor samples were not represented in the categorical location data; however, they can be identified in the map of samples by longitude and latitude (Fig. 4). The majority of the unknown samples

pertaining to categorical localities consist of ~99% of the 61,294 analyses donated from the EarthChem repository, however, these data points report precise latitude and longitude for every analysis instead.

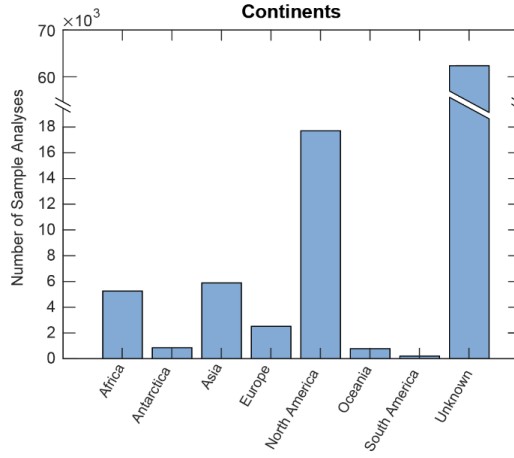

**Figure 2. Representation of all sample analyses across different continents with a break in the scale to include samples from undefined localities. In order from left to right, there are 5,263 sample analyses from Africa, 856 from Antarctica, 5,892 from Asia, 2,524 from Europe, 17,702 from North America, 786 from Oceania, 205 from South America, and 62,360 unknown sample analyses.**

The distribution of samples from different continents and countries is depicted in Fig. 2 and 3. The highest concentration of garnet analyses is located in North America with 17,702 samples, followed by Asia with 5,892 samples, Africa with 5,263 samples, and Europe with 2,524 samples (Fig. 2). The dataset contains 88 different countries of origin for garnet samples, however, only 34 countries record greater than 50 analyses each (Fig. 3). The most prominent sample countries are Canada

(5,019 sample analyses), China (1,235), India (1,426), Norway (1,288), Russia (1,544), South Africa (3,403), and the United States of America (12,489). There are 62,836 samples which do not indicate a country of origin and are listed as Unknown. It is important to note that of the 12,489 samples from the United States, 10,380 are sample analyses from Townshend Dam, Vermont (Gatewood et al., 2015), which introduces a significant bias in the dataset. It was not our intention to represent the overall natural occurrence of garnets, but rather to record the data found in the literature and list locations for samples when they were provided.

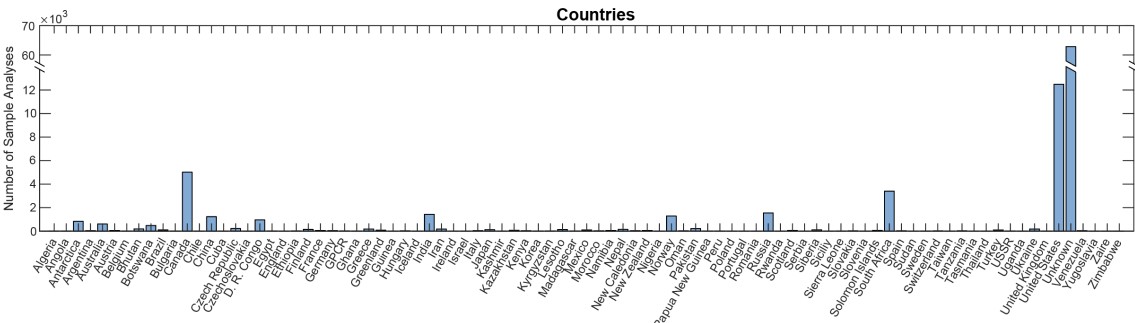


**Figure 3. Representation of all sample analyses across different countries with a break in the scale to include samples from undefined localities without obscuring the data from other, less prevalent countries. There are sample analyses from 88 total countries represented in the dataset. The most prominent sample localities are 5,019 sample analyses from Canada, 1,426 from India, 1,288 from Norway, 1,544 from Russia, 3,403 from South Africa, and 12,489 from the United States of America. There are 62,836 samples which**
**do not indicate a country of origin and are listed as unknown. Along the x-axis, D.R. Congo indicates the Democratic Republic of the Congo; GPCR is an abbreviation for sample analyses that originated from a combined location listed as Germany, Poland, and the Czech Republic; and the USSR indicates samples originating from within the historic borders of the Soviet Union.**



Earth System
Science
Data

Despite the bias towards the United States from the categorical data, there is a diverse distribution of samples around the world based on the map of longitude and latitude in Fig. 4. There are 1,786 unique locations represented by 68,364 samples (Fig. 4).

Samples originate from every major continent as well as Greenland, Iceland, New Zealand, and a handful of Pacific islands. These samples primarily originate from the EarthChem and MetPetDB repositories; however, some of the compiled peer-reviewed literature label specific longitude and latitude for each analysis, which are also included in this map (Alizai et al., 2016; Ghosh et al., 2017; Herbosch et al., 2016; Inglis et al., 2017; Javanmard et al., 2018; Kotkova and Harley, 2010; Korinevsky, 2015; Krippner et al., 2016; Manton et al., 2017; Parthasarathy et al., 1999; Patranabis-Deb, Schieber, and Basu, 2008; Schönig

et al., 2018; Sieck et al., 2019; Suwa et al., 1996). Thus, despite the bias of samples from North America, the distribution of sample localities around the world is diverse based on the reported longitude and latitude data. The distribution of sample analyses based on longitude and latitude captures the natural occurrence of garnets better than the categorical data.

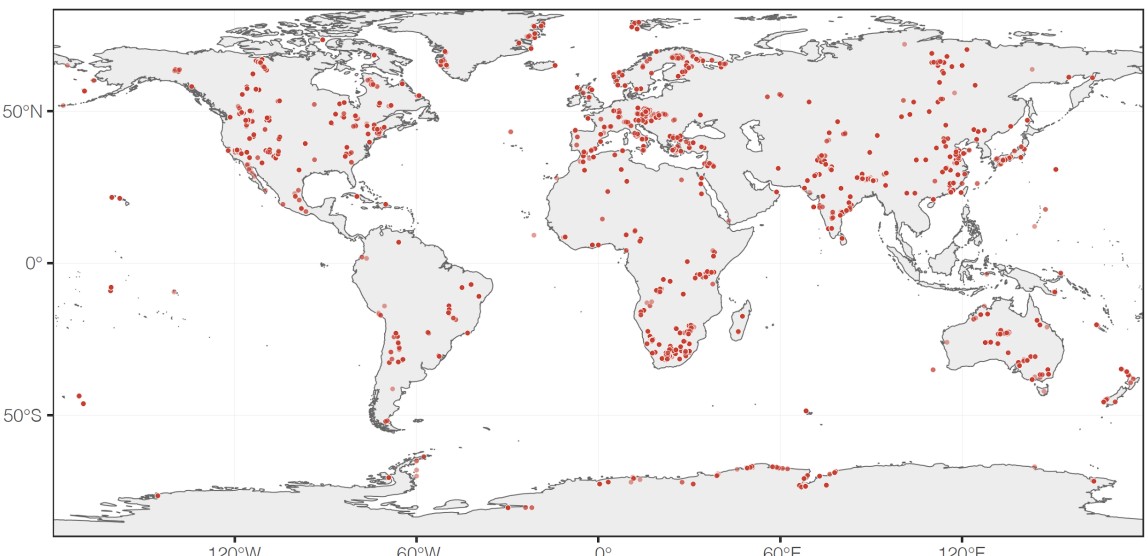

**Figure 4. A world map of the 68,364 garnet sample analyses which report longitude and latitude across 1,786 unique locations. The**
**remaining 27,224 sample analyses in the dataset do not indicate a longitude and latitude.**

### 3.3 Petrogenetic Attributes

The petrogenetic attributes (Formation, Material, Type, Composition, and Paragenesis) were chosen with increasing resolution within the dataset. Of these attributes, only 'Material,' 'Type,' 'Composition,' and 'Paragenesis' were examined further because the attribute 'Formation' contains detailed geologic descriptions taken verbatim from literature, which cannot be clustered into

specific groups, unlike the other four attributes. When only the geologic 'Formation' environment was provided, terms were determined based on descriptions from the literature and rock-type definitions from Mindat.org for each of the petrogenetic attributes. Therefore, all 95,588 sample analyses contain terms for each of the petrogenetic attributes or were recorded as unknown if unidentified. Each of the petrogenetic attributes were plotted by counts of unique categories to examine the representation of attributes within the dataset (Fig. 5). Table 3 includes an abbreviated summary of the most prominent

categories within each petrogenetic attribute and the number of sample analyses that are represented by each category. Much like the categorical locality data, the petrogenesis data should not be used to represent the overall natural occurrence of garnets.





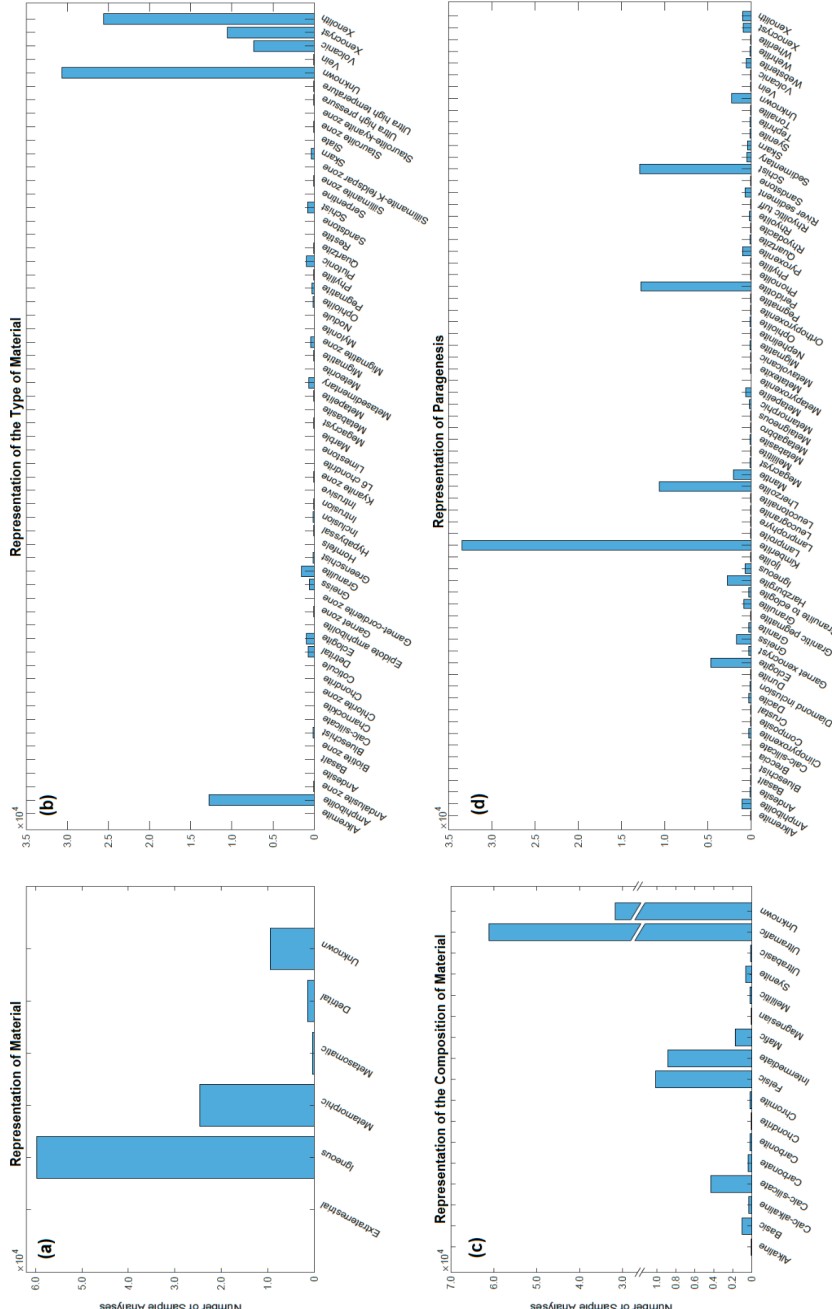

**Figure 5. Representation of petrogenetic attributes for all sample analyses in the dataset. Plots were created for the (a) parent 'Material,' (b) 'Type' of parent material, (c) 'Composition' of parent material, and (d) 'Paragenesis' of sample analyses. (a) There are six categories for Material represented by igneous, metamorphic, unknown, detrital, metasomatic, and extraterrestrial sample analyses. (b) There are 60 possible categories for the Type of parent material which are largely represented by unknown, xenolith, schist, xenocryst, and finally volcanic sample analyses. (c) There are 17 possible Compositions which are heavily biased by ultramafic and unknown compositions, followed by felsic and intermediate sample analyses. There is a break in the scale to better display the distribution of the less common compositions. (d) There are 174 categories for Paragenesis in the dataset, however, only the most prevalent 70 paragenetic modes, with counts greater than or equal to 20 analyses, were plotted to prevent cluttering the graph. The most common paragenetic modes include kimberlite, peridotite, schist, lherzolite, eclogite, and unknown sample analyses. See Table 3 for an abbreviated summary of the total number of analyses per category.**

605





| Table 3. Summary of Petrogenesis Attributes | |
| --- | --- |
| **Material** | **Number of samples** |
| Igneous | 59746 |
| Metamorphic | 24601 |
| Unknown | 9476 |
| Detrital | 1350 |
| Metasomatic | 410 |
| Extraterrestrial | 5 |
| Total Sample Analyses | 95588 |
| **Type** | **Number of samples** |
| Unknown | 30688 |
| Xenolith | 25566 |
| Amphibolite | 12751 |
| Xenocryst | 10533 |
| Volcanic | 7357 |
| Total Summary Sample Analyses | 86895 |
| **Composition** | **Number of samples** |
| Ultramafic | 61045 |
| Unknown | 31758 |
| Felsic | 1011 |
| Intermediate | 883 |
| Calc-silicate | 428 |
| Mafic | 172 |
| Total Summary Sample Analyses | 95297 |
| **Paragenesis** | **Number of samples** |
| Kimberlite | 33485 |
| Schist | 12885 |
| Peridotite | 12753 |
| Lherzolite | 10607 |
| Eclogite | 4639 |
| Harzburgite | 2748 |
| Unknown | 2262 |
| Total Summary Sample Analyses | 79379 |

Table 3. Abbreviated summary of category totals for the Petrogenetic attributes (Material, Type, Composition, Paragenesis). There are 6 total categories for the Material attribute, 60 for the Type of material, 17 possible Compositions, and finally 174 unique paragenetic modes. All of the 95,588 sample analyses have assigned categories in the dataset. The most prevalent categories and the number of sample analyses represented by each category are listed for the Type, Composition, and Paragenesis attributes for a total of 86,895, 95,297, and 79,379 analyses respectively. Plots of these attributes are depicted in Fig. 5. See the dataset in the Evolutionary System of Mineralogy Database (ESMD; http://odr.io/ESMD) for the detailed petrogenetic attributes.

610

Beginning with 'Material,' this attribute offers the lowest resolution across six categories: Extraterrestrial, Igneous,

615   Metamorphic, Metasomatic, Detrital, and Unknown (Fig. 5a). The extraterrestrial material contains garnet grains obtained from

meteorites. The igneous material (both intrusive and extrusive) consists of garnets from volcanic provinces, while the

metamorphic material contains garnets from a diverse set of metamorphic terranes due to the MetPetDB data. The metasomatic





material is dominated by skarn deposits. The detrital material consists of garnet grains found in sedimentary deposits without an associated host rock. Finally, the unknown material consists of sample analyses without any associated information. The most common parent material represented in the dataset is igneous with 59,746 analyses followed by 24,601 metamorphic, 9,476 unknown, 1350 detrital, 410 metasomatic, and 5 extraterrestrial sample analyses. (Fig. 5a; Table 3). As garnets are most commonly found within metamorphic rocks, this was an unexpected result. It is possible that the dataset may be significantly biased towards garnets of igneous origin because the samples from the EarthChem repository constitute a substantial proportion of the igneous sample analyses in the overall dataset, potentially due to the prevalence of kimberlite exploration studies.

The 'Type' of parent material is represented by 60 categories in the dataset which are plotted based on the number of samples per category in Fig. 5b. The 5 most reported material 'Types' include 30,688 unknown analyses followed by xenoliths with 25,566 analyses largely originating from EarthChem, as well as 12,751 amphibolite analyses, 10,533 xenocrysts, and finally 7,357 volcanic analyses (Table 3). These 5 categories account for ~ 91% of the overall dataset. The total number of samples for each of the other 55 types of material categories feature a substantially lower count. This is most likely a result of biases in the dataset rather than how they are represented in nature.

The 'Composition' of parent material is expressed by 17 different categories throughout the dataset (Fig. 5c). There are 61,045 ultramafic and 31,758 unknown compositions which dominate the distribution, therefore, a break in the scale is used to prevent these prevalent categories from obscuring the rest of the data (Fig. 5c; Table 3). Despite these large values, the next two most prevalent categories of composition include 1,011 felsic and 883 intermediate samples. These main compositions of the parent material account for the large number of igneous samples recorded from the EarthChem repository.

The 'Paragenesis' of sample analyses is the highest resolution attribute and presents a total 174 possible paragenetic modes of specific rock-type names derived from the literature and data repositories. We maintained as much of the terminology used to describe each sample as possible to minimize oversimplification. For example, orthogneiss and paragneiss are recorded as such rather than being lumped into the general category of gneiss. Nevertheless, some sources were more descriptive than others which created a wide range of categories in this attribute from a vague classification of igneous to a specific L6 chondrite. Paragenesis was plotted by the 70 categories which have a sample analysis count greater than or equal to 20 analyses each to visualize the most prominent paragenetic modes without cluttering the graph (Fig. 5d). This process excluded only 560 analyses from the plot because 66 categories represent less than or equal to 5 analyses each (156 analyses total) and 38 categories represent between 5 to 20 analyses each (404 analyses total). The majority of samples originate from 33,485 kimberlite analyses in the EarthChem repository, which contributes to the large number of classified igneous Material samples as well (Fig. 5d; Table 3). Other significant paragenetic modes include 12,885 schist, 12,753 peridotite, 10,607 lherzolite, 4,639 eclogite, and 2,262 unknown sample analyses (Fig. 5d; Table 3). These 6 most common paragenetic modes represent ~80% of the entire dataset. As with the other petrogenetic attributes, these data are most likely biased based on the chosen locality of these samples, the specific scientific investigation of certain studies, or the compiled literature across all data repositories and peer-reviewed literature.

## 3.4 Age, Pressure, and Temperature

The age, pressure, and temperature of sample analyses were reported in the dataset by minimum, maximum, and average values from the source literature and data repositories. Histograms for the average age, pressure, and temperature across the prevalent material types (Detrital, Igneous, Metamorphic, Metasomatic, and Unknown) were constructed with bin widths of 50 Ma, 5 kbar, and 5°C to visualize the distribution of these attributes in the dataset (Fig. 6). Multiple data sources were inconsistent with reporting the minimum, maximum, and average values. Of the 95,588 total sample analyses in the dataset, 31,479 analyses



reported the average age, 2,386 analyses reported the average pressure, and 2,399 analyses reported the average temperature. There are 6 detrital, 28,467 igneous, 1,835 metamorphic, 72 metasomatic, and 1,099 unknown analyses that report an average age (Ma) in Fig 6a. Followed by 1,750 igneous, 520 metamorphic, 12 metasomatic, and 104 unknown analyses that report an

average pressure (kbar) in Fig 6b. Finally, there are 1,843 igneous, 530 metamorphic, 12 metasomatic, and 14 unknown analyses that report an average temperature (ºC) in Fig 6c. The remaining samples in the dataset did not report average values or indicated that they were below the detection limit. It is important to note that Gatewood et al. (2015) performed a detailed analysis of Sm-Nd isotope age zoning in garnet samples, however, these ages were reported as grain zonation averages and not correlated with the 10,380 EMPA point analyses. To prevent an oversimplification and subsequent bias of garnet ages, these values were

excluded from the overall dataset.

The modes in Fig. 6 are dominated by igneous samples followed by metamorphic and unknown material. Based on the average value plots, the dataset appears heavily weighted to young igneous garnets (< 500 Ma) which formed at ≤ 80 kbar between 500 to 1500ºC (Fig. 6). The metamorphic analyses formed prior to 300 Ma at a pressure between 50 to 70 kbar and a temperature between 300 to 1300 ºC (Fig. 6). The igneous and metamorphic analyses are similar regarding their age, temperature, and

pressure of formation while the unknown samples differ in pressure, with a mode between 120 to 150 kbar. There are 12 igneous samples identified as outliers and excluded from Fig. 6b because they indicate an average pressure between 980 to 1040 kbar from Wesselton Kimberlite, South Africa (Chassé et al., 2018). These 12 kimberlite values are substantially higher than the rest of the reported pressures in the dataset (which range from 1.75 to 186.9 kbar) and were removed to prevent obscuring the distribution of sample pressure.

The age, pressure, and temperature of samples is important in regard to mineral evolution and natural kinds clustering (Boujibar et al., 2020; Hazen, 2019; Hazen and Morrison, 2020, 2021; Hazen, Morrison, and Prabu, 2020; Hazen et al., 2008, 2012, 2014; Morrison and Hazen, 2020, 2021). Episodic mineralization coinciding with supercontinent cycles has been observed across a variety of paragenetic modes and mineral species (Bradley, 2011; Hazen et al., 2012, 2014; Huston et al., 2010; Kaur and Chaudhri, 2014; Nance et al., 2014). The average garnet age maxima (100-150 Ma, 350-400 Ma, and 1500-1550 Ma in Fig. 6a)

are loosely correlated with the supercontinent formation (~430-250 Ma) and subsequent breakup (~175-65 Ma) of Pangea, however, the oldest mode of sample ages occurs during the breakup of supercontinent Columbia (~1.6-1.2 Ga; Hazen et al., 2012). The increased frequency of garnet analyses during the breakup of supercontinent cycles is of unknown origin and should be investigated further based on location and paragenetic origins.




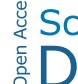



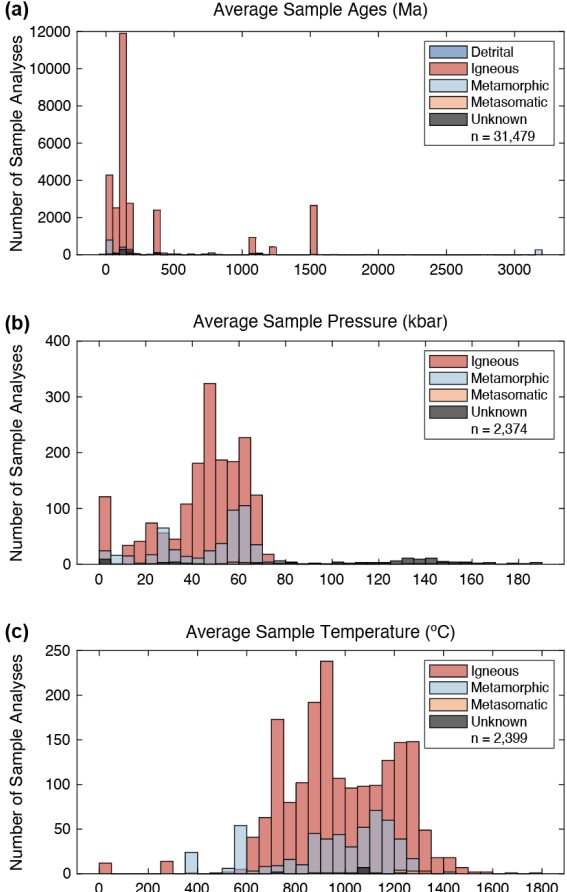

**Figure 6. Distribution of the average values of age (Ma), pressure (kbar), and temperature (ºC) reported in the dataset by material. (a) The distribution of average age with modes located at 100-150 Ma, 350-400 Ma, and 1500-1550 Ma. (b) The distribution of average pressure with a mode of igneous garnets located at 40-70 kbar, metamorphic garnets are within 50-70 kbar, and unknown analyses between 120 to 150 kbar. (c) The distribution of average temperature reveals that samples primarily form between 500 and 1500ºC with the majority of igneous, metamorphic, metasomatic, and unknown samples falling within this range. Only 31,479 analyses reported the average age (a), 2,374 analyses reported the average pressure excluding 12 kimberlite outliers (b), and 2,399 analyses reported the average temperature (c) from the entire dataset.**

### 3.5 Standardization and Data Quality

The standardization of comprehensive datasets is crucial to future research and was a primary concern throughout this project. The raw data were compiled from databases and peer-reviewed literature by manual entry or conversion to Excel with the goal of minimal manipulation while standardizing the format in which the data were represented. We ensured the quality of sample analyses through the creation of a silica confidence interval with consideration to the diversity of garnet species, which will be used as a flag to exclude questionable samples from further analysis.



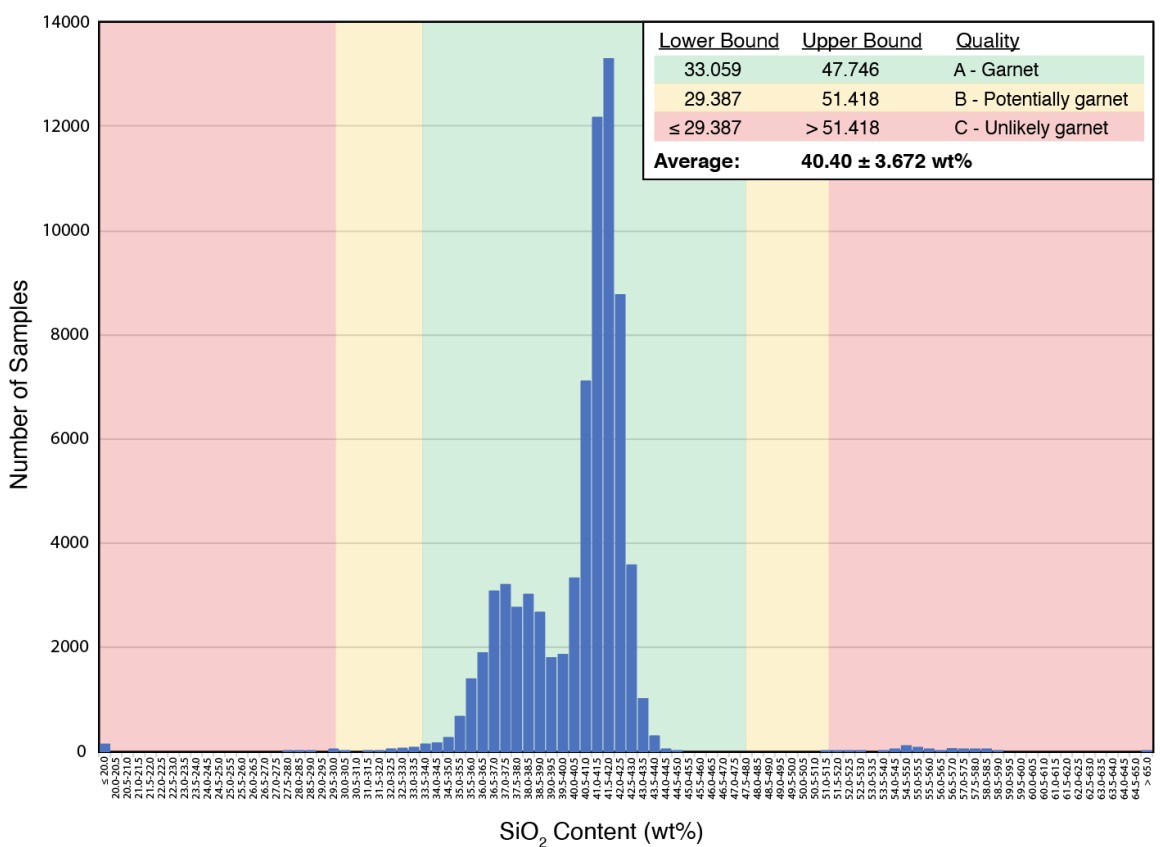

**Figure 7. Silica confidence interval indicating A, B, and C quality garnet samples for 76,018 samples that report SiO₂ wt%. The distribution is bimodal with an average of 40.40 wt% and a standard deviation of 3.672 wt%. 73,868 samples designated by 'A' are within 2 standard deviations from the mean, 639 'B' samples are within 3 standard deviations, and 1511 'C' samples are outside 3 standard deviations. A and B samples are garnets with diverse varietal species or different impurities whereas C samples are possibly not garnets. The bimodal distribution is due to the compositional difference between Mg vs. Ca-Fe²⁺-Mn garnets and Al vs. Fe³⁺-Cr-Ti garnets. Samples between 34 to 37 wt% are most likely andradite, spessartine, or uvarovite garnet species while the samples between 40 to 45 wt% are likely a high concentration of pyrope analyses.**

The 'SiO₂ Confidence Interval' attribute is used to determine the likelihood that a sample analysis is a garnet rather than an inclusion or error in the dataset. Of the total 95,588 samples, only 76,018 reported SiO₂ (wt%) while the 19,570 remaining analyses reported only trace elements or REEs. The average calculated SiO₂ content is 40.40 wt% with a standard deviation of 3.672 wt%. The bimodal distribution of silica content in Fig. 7 is largely characterized by the number of species present in the dataset, featuring one mode located with a maximum at 37.0 to 38.0 wt% and a larger mode with a maximum at 41.0 to 42.0 wt% (Fig. 7). The bimodal distribution is a consequence of the divide between Mg vs. Fe²⁺-Ca-Mn as well as Al vs. Fe³⁺-Cr garnets (Deer et al., 1982). Based on the ranges of SiO₂ content per garnet species in Table 2 adapted from Deer et al. (1982), the first mode of sample analyses between 34.0 to 38.0 wt% in Fig. 7 have a high probability of being related to almandine, spessartine, andradite and uvarovite garnet species while the second mode of samples from 40.0 to 44.0 wt% are likely pyrope dominant with grossular garnets located between each mode from 36.0 to 40.0 wt%. Nevertheless, some garnet species have



significantly different silica content, notably hydrated garnets which range from 26.0 to 38.0 wt% and majorite which reaches 50.0 to 58.0 wt% based on Table 2 (Deer et al., 1982).

| Table 2. $SiO_2$ content of common garnet species | | |
| --- | --- | --- |
| **Species** | **Ideal $SiO_2$ (wt%)** | **Range $SiO_2$ (wt%)** |
| Pyrope (Mg-Al) | 45 | 40 - 45 |
| Almandine (Fe-Al) | 35 | 36 - 38 |
| Spessartine (Mn-Al) | 35 | 35 - 38 |
| Grossular (Ca-Al) | 39 | 36 - 40 |
| Andradite (Ca-Fe) | 37 | 34 - 38 |
| Uvarovite (Ca-Cr) | 37 | 36 - 39 |
| Hydrogrossular | -- | 26 - 38 |
| Majorite | 58 | 50 - 58 |

**Table 2. Average ideal value and range of $SiO_2$ (wt%) content for the six garnet end-member species, hydrogrossular, and majorite modified from Deer et al. (1982).**

Thus, the selection of quality interval bounds must be wide enough to encompass the diversity of garnets while excluding potential errors. Picking too narrow bounding criteria could bias the dataset away from specific mineral species while picking too broad criteria could include samples or inclusions that are not garnets. Given the significance of determining sample quality, we set moderately wide intervals at 2 and 3 standard deviations from the mean. These bounds classify 73,868 'A' quality garnet samples within 2 standard deviations from the mean, followed by 639 'B' samples within 3 standard deviations, and finally 1,511 'C' samples outside 3 standard deviations (Fig. 7). There are 154 outliers with less than or equal to 20.0 wt% and 55 analyses with greater than 65.0 wt%. A group of C quality samples around 54.0 to 58.0 wt% is largely characterized by unknown garnets from EarthChem. The majorite samples from the compiled literature largely range from 38.0 to 47.0 wt% with only three sample analyses above 51.0 wt%. Additionally, only three sample analyses indicated as hydrated garnets are listed below 29.0 wt% while most hydrated samples range from ~32.0 to 38.0 wt%. Therefore, based on the distribution of samples in our dataset compared to the known ranges of garnet species in Table 2, we have included wide enough bounding criteria to account for anomalously high and low silica content in our dataset with the goal of excluding potential problematic samples from further analysis.

The raw data for the confidence interval, associated calculations, samples, and distribution of $SiO_2$ content are included in the 'SiO₂ Confidence Interval' sheet of the dataset as well as in Supplement A. This confidence interval should aid in identifying sample analyses that may be miscalculations, misidentification, a typographical error, systematic error, etc. It is important to maintain quality control of the data prior to a detailed analysis or classification of the samples. This procedure helps limit errors and allows for a separate examination of sample analyses.

### 3.6 Geochemical Analysis

Scatterplot matrices were created to analyze ten major elements found naturally within garnets (Si, $Ti^{4+}$, $Al^{3+}$, $Fe^{3+}$, $Fe^{2+}$, Mg, Ca, Na, $Mn^{2+}$, and $Cr^{3+}$). In total, five scatterplot matrices were created to represent the entire database, organized by the 'Material' attribute (Metasomatic, Detrital, Metamorphic, Igneous, and Unknown). This method was used to display the dataset so as to best compare and correlate inter-elemental relationships, and the frequencies of individual elemental weight percentages. The data is represented both visually and numerically through the scatterplot matrices and their corresponding correlation coefficients. Despite using the Kendall's Tau method, most notable for insensitivity to errors and outliers, the correlation



coefficient is still not entirely accurate. Multiple scatterplots commonly feature two or more relationships, which are not individually represented by the coefficients, as the calculation assumes a single linear relationship to be present.

### 3.6.1 Metasomatic

The metasomatic subset contains 193 samples, a majority of which originate from skarn deposits. The two strongest observable

associations occur between $Fe^{3+}$- Si and $Fe^{3+}$- $Al^{3+}$; both of which have a correlation coefficient > -0.4 (Fig. 8). Three other scatterplots ($Fe^{3+}$- $Cr^{3+}$, Ca - $Mn^{2+}$, and Na - $Cr^{3+}$) have coefficients of 0.4 or higher; however, this is calculated as an overall association drawn between two separate relationships which results in a less accurate correlation. The remaining plots have weak relationships whether they are trending negatively ($Fe^{2+}$- $Fe^{3+}$, $Fe^{2+}$- Si, $Fe^{2+}$- $Al^{3+}$, and $Ti^{4+}$- Si) or positively ($Al^{3+}$- Si and $Fe^{3+}$- $Ti^{4+}$). Most weak correlations are on account of either heteroscedasticity, outliers, disjointedness, discordant relationships, or any

combination of these.

Additionally, the Mg scatterplots have weak to negligible correlation coefficients and the data tend to clump where the highest density of Mg wt% occurs, similar to the corresponding Mg density diagram. All scatterplots analyzing Mg show a majority of their samples plotting near 0.1 wt% Mg and a smaller group of analyses plotting near 12 wt% Mg, just as the density diagram displays modes at both 0.1 wt% and 12 wt%. Elements with highly skewed density diagrams, or those with dominant modes, are

visibly reflected in their corresponding scatterplots; this occurs as well with $Mn^{2+}$ and Ca. Similar to $Mn^{2+}$, Ca, and Mg, the $Cr^{3+}$ plots display two distinct groups; however, these groups are not solely based on the modes. The first $Cr^{3+}$ group consisting of Si, $Ti^{4+}$, and $Al^{3+}$ all show a negative, disjointed, and weak relationship. While the second group consists of low $Cr^{3+}$ near 0.1 wt%, plotting widely against ranging distributions of Si, $Ti^{4+}$, and $Al^{3+}$ weight percentages. The $Cr^{3+}$ plots consisting of $Fe^{3+}$, $Fe^{2+}$, and Na only display the latter relationship of low $Cr^{3+}$ against ranging wt% distributions, as well as a few outliers. Within the

metasomatic scatterplot matrix, Na plots have near negligible relationships with the studied elements.



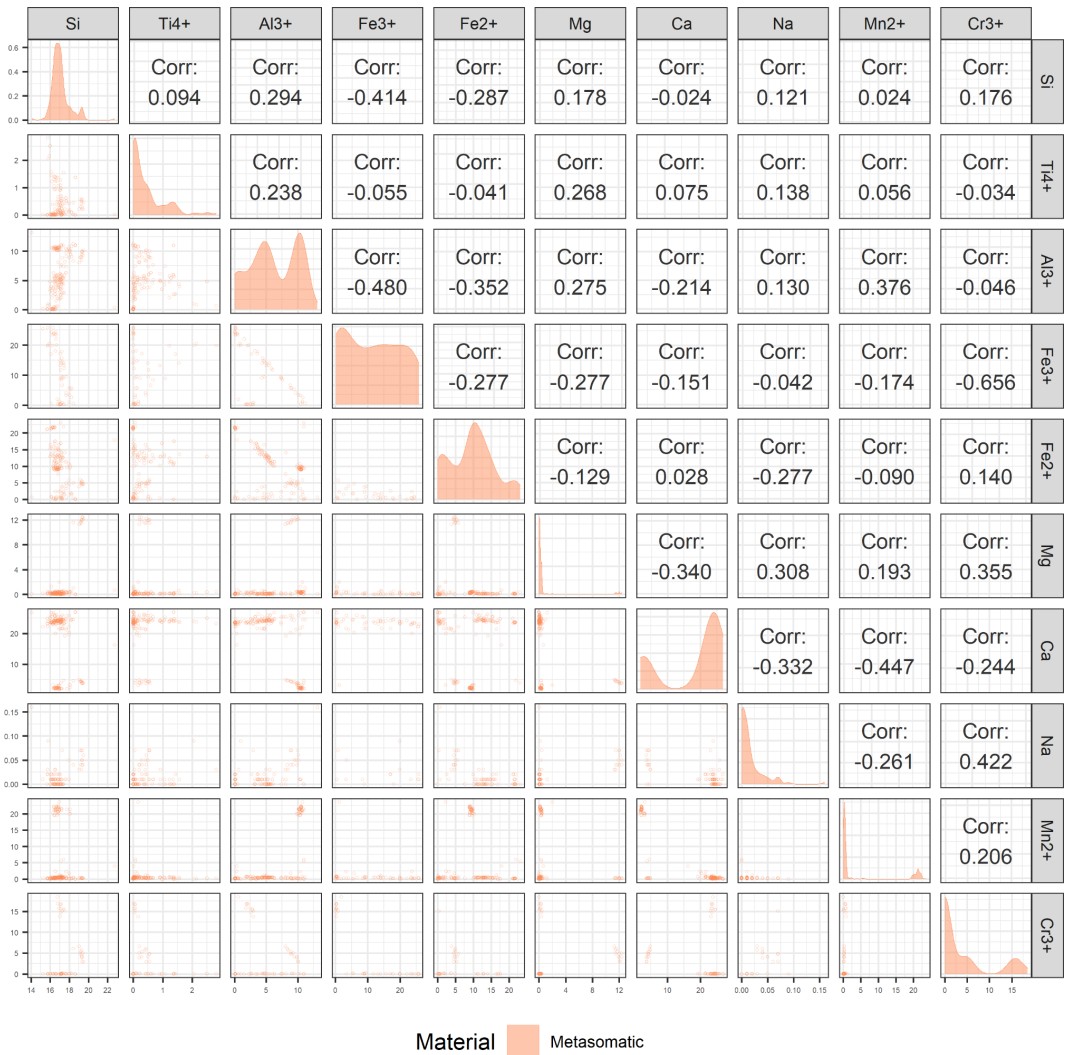

**Figure 8. Scatterplot matrix for metasomatic garnets displaying the correlation coefficients (-1 to 1) calculated by Kendall's Tau on the top. Coefficients closer to ±1 represent stronger relationships. Density diagrams in the middle, and scatterplots for the elements (wt%) on the bottom. 193 A and B quality samples are represented.**


### 3.6.2 Detrital

The detrital material consists of 747 garnet grains found in sedimentary deposits, separated from their host rock; therefore, their true petrogenesis is unknown. As a result, the description of "detrital" was retained for data analysis. Within the detrital scatterplot matrix, the strongest relationships, $Al^{3+}$ - Si and Mg - Si, both bear a positive correlation coefficient of > 0.6 (Fig. 9).

The $Cr^{3+}$- Na graph features a strong correlation coefficient of -0.59; however, this relationship consists of far fewer sample analyses. Therefore, the correlation coefficient may not be as accurate as calculated. The three plots, $Fe^{2+}$ - Si, Mg - $Fe^{2+}$, and Mg - $Al^{3+}$, all show moderate correlation coefficients of > 0.4. $Fe^{2+}$- Si and Mg - $Fe^{2+}$ are both heteroscedastic with negative slopes, whereas Mg - $Al^{3+}$ has a positive slope with a few outliers.



The remaining elemental correlation graphs all have weak to negligible relationships. In the scatterplots containing $Ti^{4+}$, $Cr^{3+}$,

$Al^{3+}$, $Fe^{3+}$, Mg, and $Mn^{2+}$, a number of analyses occur in areas where the density of each element wt% is highest. The only detrital garnet graphs depicting multiple relationships are those regarding $Mn^{2+}$. The first relationship, low $Mn^{2+}$ wt%, is shared between all the $Mn^{2+}$ graphs. This is seen comparatively with the $Mn^{2+}$ density diagram displaying a single mode at around 0.1 wt%. The second relationship can be separated further by $Mn^{2+}$ - $Fe^{2+}$ and $Mn^{2+}$ - Ca which have weak and negative correlations, while the $Mn^{2+}$ plots containing Si, $Ti^{4+}$, $Al^{3+}$, $Fe^{3+}$, and $Cr^{3+}$ reflect the modes of each element. Similar to the metasomatic

scatterplot matrix (Fig. 8), a majority of the Na detrital garnet plots have no relationships with the other elements. The only plot without a correlation coefficient is $Fe^{3+}$- Na, as there are too few analyses plotted to generate one. A majority of the detrital samples more closely resemble those of the metamorphic matrix, attributed to the similar elemental trends and modes.

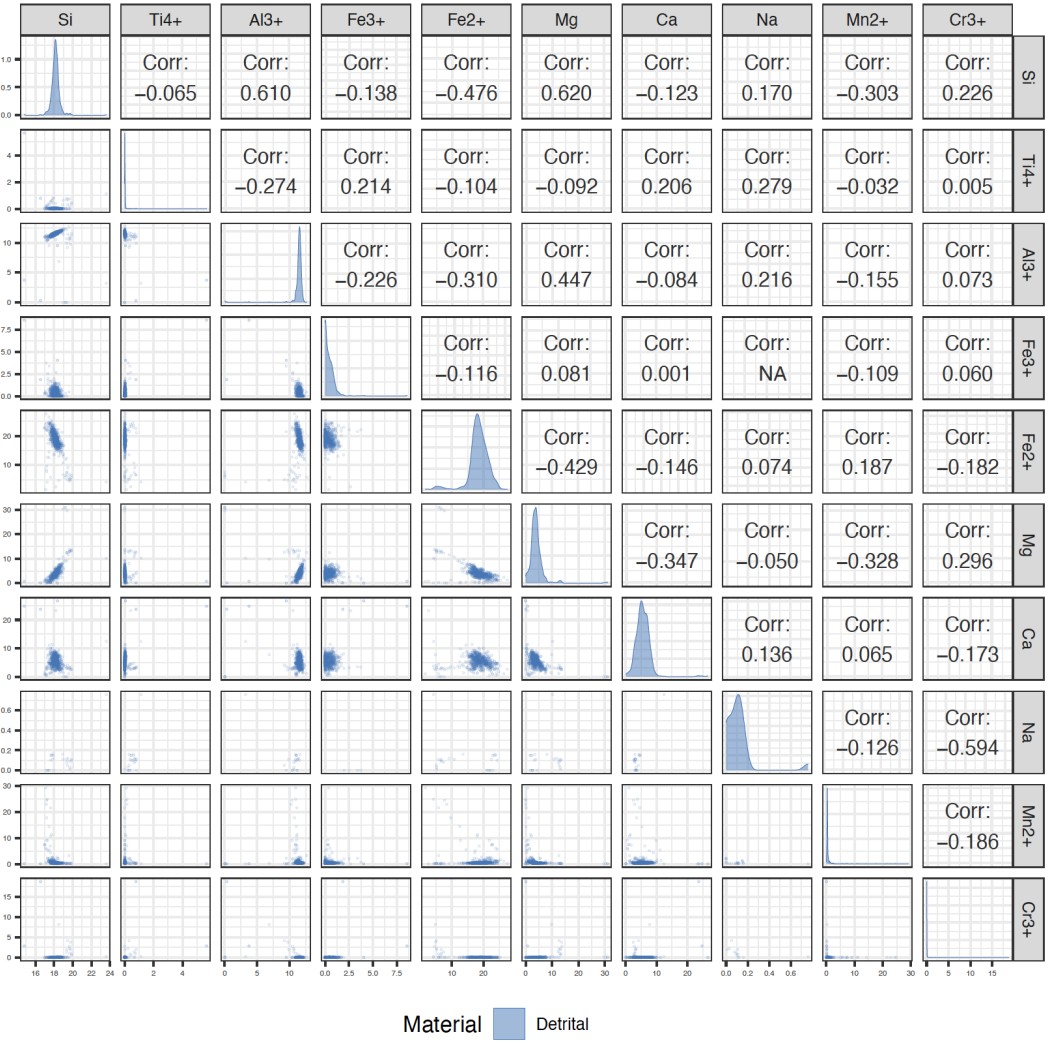

**Figure 9. Scatterplot matrix for detrital garnet grains displaying the correlation coefficients (-1 to 1) calculated by Kendall's Tau on**

**the top. Coefficients closer to ±1 represent stronger relationships. Density diagrams in the middle, and scatterplots (wt%) on the bottom. 747 A and B quality samples are represented. NA means 'not applicable' as there were too few samples to calculate a correlation coefficient.**



### 3.6.3 Metamorphic

The cleaned metamorphic material consists of 19,153 analyses with 54% originating from Gatewood et al. (2015), which consists of dominantly almandine samples from Townshend Dam, Vermont. As a result, the trends in the scatterplot matrix are biased and are not an accurate representation of naturally occurring metamorphic garnets. Bearing in mind the biases, elemental relationships with the highest correlation coefficients (~0.4 to 0.55) include $Fe^{3+}$ - $Al^{3+}$, Ca - $Fe^{2+}$, and Na - $Fe^{3+}$. The first two plots depict a moderate negative relationship (Fig. 10), whereas Na - $Fe^{3+}$ shows a moderate positive relationship. The $Mn^{2+}$ - Mg

graph bears a coefficient similar to these; however, this graph exhibits two relationships each consisting of samples low in either $Mn^{2+}$ or Mg. Many of the metamorphic scatterplots include more than one relationship per plot, much like the metasomatic matrix (Fig. 8). For example, the $Al^{3+}$ - Si plot contains two positive relationships and one weak negative relationship. Another similar graph, $Fe^{2+}$ - Si, consists of two moderate associations, one positive, one negative.

Each of the remaining scatterplots depict two relationships. The first of these is the accumulation of samples near at least one of

the respective element's modes. The second of these relationships varies between the analyzed elemental pairs. Moderate positive Mg - Si and negative Mg - $Fe^{2+}$ correlations both display an additional group near low Mg. Additionally, the $Mn^{2+}$ - $Fe^{2+}$ plot has a group of analyses with low $Mn^{2+}$ and another with a negative heteroscedastic association. The plots: Ca - Si, Ca - $Fe^{3+}$, and Ca - $Al^{3+}$, all show a group of analyses near the Ca mode of 25 wt% as well as the other element's highest wt% densities. Similar to both the metasomatic and metamorphic matrices, the Mg density diagram is skewed and has a narrow single mode,

heavily affecting a majority of the Mg scatterplots. Furthermore, the detrital garnet and metamorphic matrices display very similar density diagrams for the elements: $Ti^{4+}$, $Cr^{3+}$, $Al^{3+}$, and $Fe^{3+}$. All scatterplots, including these four elements, are affected by their modes. In contrast to the metasomatic and detrital matrices, the wt% density of Na affects all associated plots within the metamorphic matrix.







**Figure 10. Scatterplot matrix for metamorphic garnets displaying the correlation coefficients (-1 to 1) calculated by Kendall's Tau on the top. Coefficients closer to ±1 represent stronger relationships. Density diagrams in the middle, and scatterplots (wt%) on the bottom. 19,154 A and B quality samples are represented.**

### 3.6.4 Igneous

The cleaned igneous material data contains 19,451 sample analyses, the most analyses out of the five material types plotted. The plot with the strongest relationship, according to the correlation coefficient, is $Cr^{3+}$ - $Al^{3+}$ (Fig. 11). The strongest correlation trends negatively with a coefficient of -0.699 and shows very small secondary groups of analyses near the $Cr^{3+}$ and $Al^{3+}$ wt% modes. Moderate associations include $Fe^{3+}$ - Si, $Fe^{3+}$ - $Al^{3+}$, and Ca - Mg, all of which are negative and disjointed. The $Fe^{3+}$ - $Ti^{4+}$, $Fe^{2+}$ - $Fe^{3+}$, $Cr^{3+}$ - $Fe^{2+}$, and Ca - $Fe^{3+}$ plots have moderate coefficients. However, they display multiple relationships, thus altering the coefficient and producing a moderate outcome. The rest of the plots have weak to negligible relationships caused by



outliers, divided relationships, or multiple relationships. Both $Al^{3+}$ - Si and $Fe^{2+}$ - Si are good examples of weak disjointed associations with many outliers. Whereas $Fe^{2+}$ - $Al^{3+}$ has two relationships, one disjointed positive and another at the $Al^{3+}$ mode.

Much like the other matrices, the igneous matrix features many plots with multiple relationships, many of which consist of two relationships where one or more are a reflection of the modes of each element. For instance, similar to the metasomatic matrix,

all the igneous graphs that incorporate Mg show samples plotting around 12 wt% Mg. This is recognized as the dominant trend within plots such as Mg - $Al^{3+}$ and Mg - Si, both of which have secondary positive associations and outliers with low and high Mg (coinciding with the small modes on the Mg density diagram). Not all of the Mg wt% modes appear to be dominant, for example, the Mg - $Fe^{2+}$ plot has a moderate negative trend and two groups of outliers. The $Mn^{2+}$ graphs that contain Si, $Ti^{4+}$, $Al^{3+}$, Ca, and Na have two relationships influenced by each elements' modes. Additionally, the $Mn^{2+}$ - $Fe^{2+}$ graph exhibits

samples with low $Mn^{2+}$ and a second, weak, and negative association. $Mn^{2+}$ - $Cr^{3+}$ and $Mn^{2+}$ - $Fe^{3+}$ plots display an accumulation of low $Mn^{2+}$ wt%, along with many outliers. Likewise, the Ca plots have two groups of samples occurring around the 4 and 23 wt% Ca modes, similar to the metasomatic matrix. Unlike the other matrices, the igneous plots appear to have more $Cr^{3+}$ samples with a larger range of values. Two groups are observable within the $Cr^{3+}$ plots: the first appears near low $Cr^{3+}$, and the most dominant of the two, plots near the modes of the associated elements. Comparable to the metamorphic matrix (Fig. 10), the

igneous matrix's Na plots are a reflection of the 0.1 wt% Na mode. Though all the material matrices show graphs influenced by the modes of the elements, those within the igneous matrix are among the most affected by these modes, specifically: Si, $Ti^{4+}$, $Mn^{2+}$, $Fe^{2+}$, Ca, Mg, and Na plots.



**Figure 11. Scatterplot matrix for igneous garnets displaying the correlation coefficients (-1 to 1) calculated by Kendall's Tau on the top. Coefficients closer to ±1 represent stronger relationships. Density diagrams in the middle, and scatterplots (wt%) on the bottom. 19,452 A and B quality samples are represented.**

### 3.6.5 Unknown

The unknown material scatterplot matrix analyzes the rest of the 2,160 samples in the dataset. Around half of the plots (Na - $Ti^{4+}$, Ca - Mg, $Cr^{3+}$ - $Al^{3+}$, Mg - Si, and $Fe^{3+}$ - $Al^{3+}$) with the strongest correlation coefficients (~0.4 - 0.5) feature two relationships. These coefficients report weaker values, as a result of the Kendall Tau method calculating one singular linear relationship (Fig. 12). The Na - $Ti^{4+}$ scatterplots are a reflection of the low $Ti^{4+}$ (~0.1 - 1 wt%) and Na (0.1 wt%) modes. Both Ca - Mg and $Cr^{3+}$ - $Al^{3+}$ have negative relationships (the former weak, the latter moderate) with a secondary group of samples plotting near low Mg or $Cr^{3+}$ wt%. The Mg - Si plot features a number of low Mg samples as well as a positive relationship, where a majority of





samples plot around 12 wt% Mg. This mode of 12 wt%, within the unknown matrix, is also seen in both the metasomatic and igneous matrices. $Fe^{3+}$ - $Al^{3+}$ has a moderate to weak coefficient of -0.390; the graph displays the two relationships, one negative and disjointed, while the other comprises a group of analyses with low $Fe^{3+}$. The $Fe^{3+}$ - Si graph only depicts one moderate, negative, and disjointed relationship with a coefficient of -0.400.

The rest of the plots have weak to negligible correlation coefficients influenced by outliers, disjointedness, or multiple
relationships. For instance, Mg - $Fe^{2+}$ and $Fe^{2+}$ - Si both have negative heteroscedastic moderate relationships with secondary groups plotting near low Mg or $Fe^{2+}$ wt%. Furthermore, $Mn^{2+}$ - Ca and $Cr^{3+}$ - Si have weak negative relationships, and groups of analyses showing low $Mn^{2+}$ or Si modes. The $Al^{3+}$ - Si graph has two moderate relationships, one positive, one negative that intersect at each elements' modes. The curved nature of the $Fe^{2+}$ - $Al^{3+}$ plot, like most relationships, is a reflection of both elements' modes. Likewise, the Mg - $Al^{3+}$ graph displays a curved shape, similar to the same plot in the igneous matrix. The three
parts making up the curve of this graph are all plotted modes, two Mg modes (~0.1 and 12 wt%) and one $Al^{3+}$ mode (~11 wt%). Unlike the other matrices, the unknown graphs have more Na samples with a larger range of values. A majority of these scatterplots are influenced by the 0.1 wt% Na mode. Na - Si and Na - Ca have an additional weak to negligible positive association. Similar to the igneous, metamorphic, and metasomatic matrices, the unknown Ca graphs have two groups of samples occurring around 4 and 24 wt% Ca. The unknown matrix is greatly affected by the modes of each element, specifically the $Ti^{4+}$,
$Mn^{2+}$, Na, $Fe^{3+}$, and Ca plots. A majority of the unknown samples most closely resemble those of the igneous matrix, attributed to the similar elemental trends and modes.





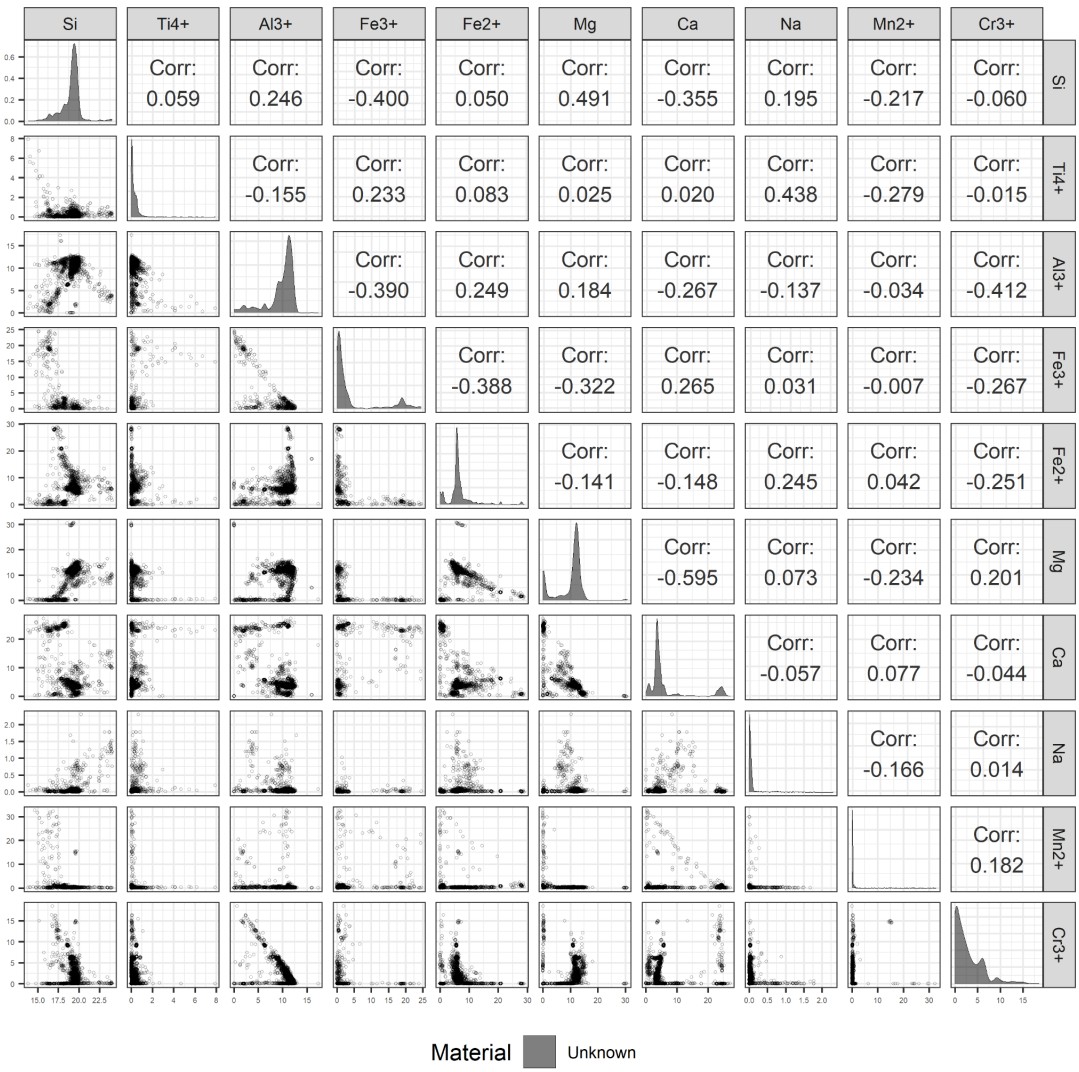

**Figure 12. Scatterplot matrix for unknown garnets displaying the correlation coefficients (-1 to 1) calculated by Kendall's Tau on the top. Coefficients closer to ±1 represent stronger relationships. Density diagrams in the middle, and scatterplots (wt%) on the bottom. 2,161 A and B quality samples are represented.**

### 3.6.6 Comparisons of Geochemical Analyses

A majority of the relationships and trends within the scatterplot matrices are a direct result of the density distributions and modes of elemental values. Some of the elemental modes do not differ between material types, such as Si, which is expected. All of the analyses across each material type have a Si range of around 15 to 20 wt%. Similarly, both the Na and $Ti^{4+}$ diagrams have distributions near 0.1 wt% across the matrices, with slight variations. The $Mn^{2+}$ and $Cr^{3+}$ plots also have samples that group near 0.1 wt%, however, the distribution varies across a few material types. $Cr^{3+}$ has a wide range within the igneous diagram (0.1 - 9 wt%) and even more so in the metasomatic and unknown diagrams (0.1 - 20 wt%). The $Mn^{2+}$ graphs range within the metamorphic plot, from 0.1 - 9 wt%, and display a secondary mode in the metasomatic plot at 23 wt%.



The remaining elemental density diagrams have wt% distributions and modes that differ between material matrices. The metasomatic, igneous, and unknown matrices have ranges between 0.1 - 25 $Fe^{3+}$ wt% and are weakly bimodal. Meanwhile, the detrital and metamorphic matrices have significantly smaller ranges (0.1 - 2.5 $Fe^{3+}$ wt%) with a mode around 0.1 wt%. The detrital, igneous, and unknown diagrams display a mode at 12 $Al^{3+}$ wt%, the metamorphic diagrams show one mode at 11 wt% and the metasomatic diagrams show two modes at 4.8 and 10.5 wt%. Both the metamorphic and metasomatic $Fe^{2+}$ diagrams are

multimodal, however, the metamorphic matrix shows the largest range, from 0.1 - 30 wt%, as well as highest mode at 25 wt%. The igneous $Fe^{2+}$ graph has a normal distribution, and the smallest range of 4 - 9 wt% with a mode at 5 wt%. Unknown and metasomatic graphs show two modes around 0.1 and 12 Mg wt%; the unknown plot has the highest range from 0.1 - 18 wt%. The detrital diagram displays a mode around 4 Mg wt%, the metamorphic diagram shows one around 1 wt%, and the igneous diagram shows a mode around 12 wt%. The final element, Ca, is represented by a bimodal distribution in the metasomatic graph

with two modes at 3 and 24 wt%, and a multimodal distribution in the unknown matrix with modes around 1, 4, and 24 wt%. Additionally, the igneous Ca diagram displays one main mode at 4 wt%. The detrital diagram shows a main mode at 5 wt%, and the metamorphic graph shows two main modes at 1 and 5 wt%.

### 3.6.7 Relationships Between Geochemical Material and Garnet Species

The scatterplot matrix associations are heavily influenced by the elemental modes seen in the density diagrams, for this reason,

two or more relationships frequently occur within each scatterplot. These relationships are also seen within the pyralspite and ugrandite solid solution series, connecting the six major garnet species: almandine, grossular, pyrope, andradite, spessartine, uvarovite. Additional scatterplot relationships are associated with the minor garnet species, majorite and calderite.

The majority of the known garnet samples within the metasomatic analysis include andradite, grossular, spessartine, and uvarovite. The stronger metasomatic relationships, $Fe^{3+}$ - $Al^{3+}$, $Fe^{3+}$ - $Cr^{3+}$, Ca - $Mn^{2+}$, are reflected in the pyralspite and

ugrandite solid solution series. The relationship $Fe^{3+}$ - $Al^{3+}$ represents the series between andradite and grossular; $Fe^{3+}$ - $Cr^{3+}$, the andradite-uvarovite series; and Ca - $Mn^{2+}$, the grossular-spessartine series.

Known detrital garnet analyses are represented by almandines, spessartines, and pyropes. The strong relationship, Mg - $Fe^{2+}$, is the solid solution between pyrope and almandine. Weak to moderate detrital associations include $Mn^{2+}$ - $Fe^{2+}$, the spessartine-almandine series, and $Mn^{2+}$ - Ca, the spessartine-grossular series.

Within the metamorphic matrix, the strongest relationships include: $Fe^{3+}$ - $Al^{3+}$, the andradite-grossular series; Ca - $Fe^{2+}$, the grossular-almandine series; and $Mn^{2+}$ - Mg, the spessartine-pyrope series. Some of the moderate to weak associations include: Mg - Si which may be caused by the majorite analyses, Mg - $Fe^{2+}$ the pyrope-almandine series, and $Mn^{2+}$ - Ca, the spessartine-grossular series, which may also be a result of influence from the calderite garnets. Bias within the metamorphic matrix is present, as more than half of the samples are almandines originating from one source, Townshend Dam, Vermont. Therefore, this

matrix does not accurately represent all naturally occurring metamorphic garnets.

Throughout the igneous matrix, andradite, almandine, spessartine, and pyrope make up the majority of the known igneous analyses. This matrix also includes majorite, represented by the relationship between Mg - Si. The stronger relationships include: $Cr^{3+}$ - $Al^{3+}$, the solid solution series for uvarovite-grossular; $Fe^{3+}$ - $Al^{3+}$, the andradite-grossular series; Ca - Mg, the grossular-pyrope series; and $Mn^{2+}$ - $Fe^{2+}$, the spessartine-almandine series.

The unknown matrix includes strong relationships of: Ca - Mg, $Cr^{3+}$ - $Al^{3+}$, Mg - Si, and $Fe^{3+}$ - $Al^{3+}$. The Mg - Si relationship represents majorite garnets, the Ca - Mg relationship represents the grossular-pyrope series, the $Cr^{3+}$ - $Al^{3+}$ relationship represents the uvarovite-grossular series, and $Fe^{3+}$ - $Al^{3+}$ relationship represents the andradite-grossular solid solution series. Additionally, there is a moderate relationship between $Mn^{2+}$ - Ca, the series between spessartine and grossular. The detrital and unknown





matrices consist of samples with undetermined petrogeneses. Comparing these analyses and their relationships within the matrices, the majority of the unknown samples align with igneous trends, while the majority of the detrital samples are likely to be of metamorphic origin.

**4 Future Work**

Future work with cluster analysis will focus on dividing garnet samples into different groups that correspond to their paragenetic modes (such as igneous or metamorphic types), formational environment (different tectonic settings), or temperature-pressure

conditions which is consistent with natural kinds clustering. For example, pyrope is known to occur in mantle-derived ultramafic rocks, including eclogite and kimberlite, as well as in amphibole and biotite schists (Deer et al., 1982). Similarly, andradite is frequently encountered in both contact metamorphic environments as well as in alkali igneous rocks. We suggest that cluster analysis will reveal discrete combinations of compositions and other attributes for these contrasting igneous and metamorphic parageneses for pyrope and andradite. Compared with defining garnet groups based on chemical compositions, these future paths

might have further implications for understanding the formation of the garnets, identifying source lithologies for detrital garnets, and documenting the co-evolution of garnet with Earth's environment.

This database aims to incorporate future studies and sample analyses, after publication, in the Evolutionary System of Mineralogy Database (ESMD). Ultimately, we intend to develop a system in which researchers can upload their samples to this database for continuous documentation and expansion of garnet mineralogical data.

**5 Data Availability**

These data are freely available from the Evolutionary System of Mineralogy Database (ESMD; https://odr.io/ESMD-Garnet). https://doi.org/10.48484/camh-xy98 (Chiama et al., 2022).

**6 Conclusion**

In a society increasingly dependent on the internet and open-access data resources, it is imperative to maintain the accessibility,

reproducibility, and interoperability of data in accordance with the FAIR guiding principles. Thus, the data science goals of this study were to record dark data for garnet group minerals in a standardized format that is readily accessible and to combine those dark data with current databases, which facilitates the access to valuable scientific information while continuing to expand the availability of mineralogical data for future studies. We encourage scientists to contribute to these large and growing data repositories of mineralogical information, which are proving invaluable in the advancement of scientific discovery.

**Supplemental Data:**

Supplement A: Calculations and data used to create the $SiO_2$ confidence interval.
Supplement B: A detailed analysis of the 275 original EMPA point-analyses performed for the dataset.
Supplement C: A summary of the average oxide totals for the 275 original EMPA point-analyses.





## Author Contributions

R.M.H. conceptualized the project idea. R.M.H., S.Z., S.M., E.S.B. A.B., and J.A.N. mentored and provided advice. R.M.H., J.A.N., M.W., K.L., and F.S. provided resources of garnet samples. K.C., M.G., I.L., and R.R. performed data curation, development of dataset methodology, formal analysis of data, investigation, and manuscript preparation and finalization. K.C., M.G., I.L., R.R., E.S.B. and A.B. prepared samples and analyzed their compositions. K.C. and S. Z. created visualizations for individual attributes in the dataset. K.C. and I.L. performed investigations for and created visualizations of the silica confidence

interval and scatterplot matrices respectively. R.M.H., J.A.N., S.Z., S.M., A.B., M.W., K.L., and F.S reviewed and edited the manuscript.

## Competing Interests

The authors declare that they have no conflict of interest.

## Acknowledgements

We thank the Carnegie Institution for Science's Earth and Planets Laboratory and the 4D Deep-Time Data Driven Initiative (4d.carnegiescience.edu) for supporting us during this research project. We thank Julia A. Nord and Robert M. Hazen for their donation of garnet samples for EMPA analysis as well as Kerstin Lehnert and Frank Spear for their donation of garnet sample analyses included in the EarthChem and MetPetDB repositories respectively. We also thank Michael Walter, the Deputy Director of the Earth and Planets Laboratory, for donating his personal dataset of majorite samples for this project. We would like to

thank Emma Bullock for arranging time to operate the SEM and EMPA to evaluate our samples as well as for advice and guidance with sample analyses, Guiseppina Kysar for advice and feedback on the manuscript, Rebecca Schmidt for help with compiling garnet analyses from the literature, Matthew Endries for Rcode advice and edits, as well as Michael Naylor Hudgins for manuscript advice and figure preparation. Funding for this project was provided, in part, by the NASA Astrobiology Institute (Cycle 8) ENIGMA: Evolution of Nanomachines In Geospheres and Microbial Ancestors (80NSSC18M0093) and the John

Templeton Foundation.

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
