# Peer review of "The secret life of garnets: A comprehensive, standardized dataset of garnet geochemical analyses integrating localities and petrogenesis."

_Earth System Science Data, 2023_

## Author Comment (AC1)

List of references in alphabetical order excluding MetPetDB & Earthchem:

1. Alizai et al. (2016)
2. Altunbey and Kiliç, (2019)
3. Antao (2013)
4. Antao and Cruickshank (2018)
5. Antao and Round (2014)
6. Armbruster et al. (1998)
7. Beard and Drake (2007)
8. Bell et al. (1995)
9. Bónová et al. (2018)
10. Bulanova et al., 2010
11. Bulanova, Unpub
12. Capetown, Unpubl
13. Cenki-Tok and Chopin (2006)
14. Chasse
15. Chen et al. (1996)
16. Chen et al. (2004)
17. Chen et al. (2015)
18. Dasgupta et al. (1987)
19. Davies et al., 1999
20. Davies et al., 2004B
21. Deer, Howie, and Zussman (1982)
22. Dzigbodi-Adjimah (2004)
23. EarthChem
24. Enami et al. (1993)
25. Exley (1982)
26. Fedorowich et al. (1993)
27. Gadas et al. (2012)
28. Galuskina et al. (2010)
29. Galuskina et al. (2010)
30. Gaspar et al. (2008)
31. Gatewood et al., 2015
32. Ghosh and Morishita (2011)
33. Ghosh et al. (2017)
34. Gurney and Moore, 1986
35. Harangi et al. (2001)
36. Harte and Cayzer, 2007
37. Haynes et al. (2003)
38. Hilton (2000)
39. Hode Vuorinen et al. (2005)
40. Huang et al. (2020)
41. Hutchison, 1997
42. Inglis et al. (2017)
43. Ivanova et al., 2017
44. Jamtveit et al. (1997)
45. Javanmard et al. (2018)
46. Jeong and Kim (1999)
47. Kaminsky et al., 2001
48. Katerinopoulou (2009)
49. Kawakami et al. (2019)
50. Kebede, Koeberl, and Koller (2001)

51.     Korinevsky (2015)
52.     Kotkova and Harley (2010)
53.     Krippner et al. (2016)
54.     Li et al. (2018)
55.     Locock (2008)
56.     Makrygina and Suvorova (2011)
57.     Manton et al. (2017)
58.     Marks et al. (2008)
59.     Mason (1966)
60.     Mcaloon and Hofmeister (1995)
61.     MetPetDB
62.     Meyer and Mahin, 1986
63.     Meyer and Svisero, 1975
64.     Moore and Gurney, 1985
65.     Moore and Gurney., 1989
66.     Moore et al., 1991
67.     Mueller and Delor (1991)
68.     Munno et al. (1980)
69.     Naimo et al. (2002)
70.     Naimo et al. (2003)
71.     Nixon et al. (1963)
72.     Padovani and Tracy (1981)
73.     Pal and Das (2010)
74.     Parthasarathy et al. (1999)
75.     Patranabis-Deb, Schieber, and Basu (2008)
76.     Philpotts et al. (1972)
77.     Plümper et al. (2014)
78.     Pokhilenko et al., 2004
79.     Preston et al. (2001)
80.     Pribavkin, Avdonina, and Zamyatin (2012)
81.     Quartieri et al. (2002)
82.     RRUFF
83.     Russell et al. (1999)
84.     Salnikova et al. (2019)
85.     Schertl et al. (2018)
86.     Schingaro et al. (2001)
87.     Schingaro et al. (2016)
88.     Schönig et al. (2018)
89.     Sibi and Subodh (2017)
90.     Sieck et al. (2019)
91.     Sipahi et al. (2017)
92.     Sobolev et al. (2011)
93.     Sobolev et al., 1997
94.     Sobolev et al., 2004
95.     Song et al. (2003)
96.     Stachel and Harris, 1997
97.     Stachel et al., 1998
98.     Stachel et al., 2000
99.     Stähle et al., 2011
100.    Stalder and Rozendaal (2005)
101.    Suwa et al. (1996)

102.    Tappert et al., 2005B
103.    Taran and Langer (2000)
104.    Thomson et al. (2016)
105.    Thomson, Unpub
106.    Tian et al. (2019)
107.    Tsai et al., 1979
108.    Volkova et al. (2014)
109.    Von Knorring et al. (1986)
110.    Wang et al. (1999)
111.    Wang et al., 2000
112.    Wang, Essene, and Zhang (2000)
113.    Weiss (1949)
114.    Wilding, 1990
115.    Yang et al. (2013)
116.    Zedgenizov et al., 2014
117.    Zeh and Gerdes (2014)
118.    Zhang et al. (2022)
119.    Zhou et al. (2017)

(Weiss, 1949; Nixon et al., 1963; Mason, 1966; Gurney and Moore, 1986; Fedorowich et al., 1995; Bell et al., 1995; Wang et al., 2000, 2000; Preston et al., 2002; Song et al., 2003; Galuskina et al., 2010a, 2010b; Sobolev et al., 2011; Krippner et al., 2016; Sipahi, Ferkan et al., 2017; Li et al., 2018a; Bonova et al., 2018; Kawakami et al., 2019; Salnikova et al., 2019; Tian et al., 2019)(Deer et al., 2013)

(Davies et al., 2004)(Philpotts et al., 1972; Wilding, 1990; Hutchison, 1997; Stachel et al., 1998; Stachel, T. et al., 2000; Kaminsky et al., 2001; Sobolev et al., 2004; Bulanova et al., 2010; Huang et al., 2020; Zhang et al., 2022)

(Cenki-Tok and Chopin, 2006; Antao and Round, 2014)(Antao and Cruickshank, 2018)(Reinecke et al., 1985; Yang et al., 2013; Chen et al., 2015; Rahmani Javanmard et al., 2018)

(Alizai et al., 2016; Zhou et al., 2017)(Exley, 1982; Armbruster, 1998; Russell et al., 1999; Hilton, 2000; Taran and Langer, 2000; Quartieri et al., 2002; Naimo et al., 2003; Dzigbodi-Adjimah, 2004; Hode Vuorinen et al., 2005; Marks et al., 2008; Katerinopoulou, 2009; Makrygina and Suvorova, 2011; Antao, 2013; Plümper et al., 2014; Korinevsky, 2015; Ghosh et al., 2017; Li et al., 2018a; Schönig et al., 2018)

(Munno et al., 1980; von Knorring et al., 1986; Dasgupta et al., 1987; Mueller and Delor, 1991; Enami et al., 1993; McAloon and Hofmeister, 1995; Chen et al., 1996, 2004; Suwa et al., 1996; Jamtveit et al., 1997; Jeong and Kim, 1999; Wang et al., 1999; Parthasarathy et al., 1999; Schingaro et al., 2001; Harangi, 2001; Haynes et al., 2003; Naimo et al., 2003; Walter et al., 2004; Stalder and Rozendaal, 2005; Beard and Drake, 2007; Gaspar et al., 2008; Locock, 2008; Pal and Das, 2010; Kotková and Harley, 2010; Ghosh and Morishita, 2011; Zeh and Gerdes, 2014; Schingaro, 2016; Thomson et al., 2016; Schertl et al., 2018)(Ivanova et al., 2017; Altunbey and Kiliç, 2019)(Chassé et al., 2018)

(Padovani and Tracy, 1981; Fujimaki et al., 1984; Kebede et al., 2001; Patranabis-Deb et al., 2009; Pribavkin et al., 2013; Gadas et al., 2013; Plümper et al., 2014; Inglis et al., 2017; Manton et al., 2017; Sibi and Subodh, 2017; Li et al., 2018b; Sieck et al., 2019)(Stähle et al., 2011; Zedgenizov et al., 2014; Volkova et al., 2014; Sipahi et al., 2017)

References

Alizai, A., Clift, P.D., and Still, J., 2016, Indus Basin sediment provenance constrained using garnet geochemistry: Journal of Asian Earth Sciences, v. 126, p. 29–57, doi:10.1016/j.jseaes.2016.05.023.

Altunbey, M., and Kiliç, A.D., 2019, Fluid inclusion and oxygen isotope studies in garnets related to Çavuşlu Skarn iron mineralization, East Turkey: Journal of African Earth Sciences, v. 149, p. 465–473, doi:10.1016/j.jafrearsci.2018.09.004.

Antao, S.M., 2013, Three cubic phases intergrown in a birefringent andradite-grossular garnet and their implications: Physics and Chemistry of Minerals, v. 40, p. 705–716, doi:10.1007/s00269-013-0606-4.

Antao, S.M., and Cruickshank, L.A., 2018, Crystal structure refinements of tetragonal (OH,F)-rich spessartine and henritermierite garnets: Acta Crystallographica Section B Structural Science, Crystal Engineering and Materials, v. 74, p. 104–114, doi:10.1107/S2052520617018248.

Antao, S.M., and Round, S.A., 2014, Crystal chemistry of birefringent spessartine: Powder Diffraction, v. 29, p. 233–240, doi:10.1017/S0885715614000062.

Armbruster, 1998, Crystal chemistry of Ti-bearing andradites: European Journal of Mineralogy, v. 10, p. 907–922, doi:10.1127/ejm/10/5/0907.

Barbosa, E.S.R., Brod, J.A., Junqueira-Brod, T.C., Dantas, E.L., Cordeiro, P.F. de O., and Gomide, C.S., 2012, Bebedourite from its type area (Salitre I complex): A key petrogenetic series in the Late-Cretaceous Alto Paranaíba kamafugite–carbonatite–phoscorite association, Central Brazil: Lithos, v. 144–145, p. 56–72, doi:10.1016/j.lithos.2012.04.013.

Beard, D., and Drake, M., 2007, A melilite-bearing high-temperature calcic skarn ...: mafiadoc.com, https://mafiadoc.com/a-melilite-bearing-high-temperature-calcic-skarn-_5baa1c57097c47c5688b461c.html (accessed February 2019).

Bell, D.R., Ihinger, P.D., and Rossman, G.R., 1995, Quantitative analysis of trace OH in garnet and pyroxenes: American Mineralogist, v. 80, p. 465–474, doi:10.2138/am-1995-5-607.

Bonova, K., Mikuš, T., and Bóna, J., 2018, Is Cr-Spinel Geochemistry Enough for Solving the Provenance Dilemma? Case Study from the Palaeogene Sandstones of the Western Carpathians (Eastern Slovakia): Minerals, v. 8, p. 543, doi:10.3390/min8120543.

Bulanova, G., Walter, M., Smith, C., Kohn, S., Armstrong, L., Blundy, J., and Gobbo, L., 2010, Mineral inclusions in sublithospheric diamonds from Collier 4 kimberlite pipe, Juina, Brazil: Subducted protoliths, carbonated melts and primary kimberlite magmatism: Contributions to Mineralogy and Petrology, v. 160, p. 489–510, doi:10.1007/s00410-010-0490-6.

Cenki-Tok, B., and Chopin, C., 2006, Coexisting calderite and spessartine garnets in eclogite-facies metacherts of the Western Alps: Mineralogy and Petrology, v. 88, p. 47–68, doi:10.1007/s00710-006-0146-4.

Chassé, M., Griffin, W.L., Alard, O., O'Reilly, S.Y., and Calas, G., 2018, Insights into the mantle geochemistry of scandium from a meta-analysis of garnet data: Lithos, v. 310–311, p. 409–421, doi:10.1016/j.lithos.2018.03.026.

Chen, M., Sharp, T.G., El Goresy, A., Wopenka, B., and Xie, X., 1996, The Majorite-Pyrope + Magnesiowüstite Assemblage: Constraints on the History of Shock Veins in Chondrites: Science, v. 271, p. 1570–1573, doi:10.1126/science.271.5255.1570.

Chen, M., Xie, X., and Goresy, A.E., 2004, A shock-produced (Mg, Fe)SiO3 glass in the Suizhou meteorite: Meteoritics & Planetary Science, v. 39, p. 1797–1808, doi:10.1111/j.1945-5100.2004.tb00076.x.

Chen, Y.-X., Zhou, K., Zheng, Y.-F., Chen, R.-X., and Hu, Z., 2015, Garnet geochemistry records the action of metamorphic fluids in ultrahigh-pressure dioritic gneiss from the Sulu orogen: Chemical Geology, v. 398, p. 46–60, doi:10.1016/j.chemgeo.2015.01.021.

Chiama, K., Gabor, M., Rutledge, R., Lupini, I., Nord, J. A., Zhang, S., Boujibar, A., Spear, F., Morrison, S. M., Hazen, R. M., 2021, Garnet mineral geochemistry data download from the MetPetDB (re3data.org), August 2019, Version 1.0. Interdisciplinary Earth Data Alliance (IEDA). https://doi.org/10.26022/IEDA/112173. Accessed 2021-10-29.

Chiama, K., Gabor, M., Nord, J. A., Lupini, I., Boujibar, A., Morrison, S. M., Lehnert, K., Rutledge, R., Zhang, S., Hazen, R. M., 2021, Garnet mineral geochemistry data download from the EarthChem Portal, August 2019, Version 1.0. Interdisciplinary Earth Data Alliance (IEDA). https://doi.org/10.26022/IEDA/112171. Accessed 2021-10-27.

Dasgupta, S., Bhattacharya, P.K., Banerjee, H., Fukuoka, M., Majumdar, N., and Roy, S., 1987, Calderite-rich garnets from metamorphosed manganese silicate rocks of the Sausar Group, India, and their derivation: Mineralogical Magazine, v. 51, p. 577–583, doi:10.1180/minmag.1987.051.362.12.

Davies, R.M., Griffin, W.L., O'Reilly, S.Y., and McCandless, T.E., 2004, Inclusions in diamonds from the K14 and K10 kimberlites, Buffalo Hills, Alberta, Canada: diamond growth in a plume? Lithos, v. 77, p. 99–111, doi:10.1016/j.lithos.2004.04.008.

Deer, W.A., FRS, Howie, R.A., and Zussman, J., 2013, An Introduction to the Rock-Forming Minerals: Mineralogical Society of Great Britain and Ireland, doi:10.1180/DHZ.

Dzigbodi-Adjimah, K., 2004, The mineralogy and petrography of the ferruginous manganese rocks at Mankwadzi, Ghana: Journal of African Earth Sciences, v. 38, p. 293–315, doi:10.1016/j.jafrearsci.2003.08.001.

Enami, M., Zhao, Z., and Wang, Q., 1993, A calderitic garnet paragenesis in granitic gneisses in the Su–Lu ultra high-pressure terrane, eastern China: Mineralogical Journal, v. 16, p. 268–277, doi:10.2465/minerj.16.268.

Exley, R.A., 1982, Electron microprobe studies of Iceland Research Drilling Project high-temperature hydrothermal mineral geochemistry: Journal of Geophysical Research: Solid Earth, v. 87, p. 6547–6557, doi:10.1029/JB087iB08p06547.

Fedorowich, J.S., Jain, J.C., Kerrich, R., and Sopuck, V., 1995, Trace-element analysis of garnet by laser-ablation microprobe ICP-MS.

Fujimaki, H., Tatsumoto, M., and Aoki, K., 1984, Partition coefficients of Hf, Zr, and ree between phenocrysts and groundmasses: Journal of Geophysical Research: Solid Earth, v. 89, p. B662–B672, doi:10.1029/JB089iS02p0B662.

Gadas, P., Novák, M., Talla, D., and Vašinová Galiová, M., 2013, Compositional evolution of grossular garnet from leucotonalitic pegmatite at Ruda nad Moravou, Czech Republic; a complex EMPA, LA-ICP-MS, IR and CL study: Mineralogy & Petrology, v. 107, p. 311–326, doi:10.1007/s00710-012-0232-8.

Galuskina, I.O. et al., 2010a, Bitikleite-(SnAl) and bitikleite-(ZrFe): New garnets from xenoliths of the Upper Chegem volcanic structure, Kabardino-Balkaria, Northern Caucasus, Russia: American Mineralogist, v. 95, p. 959–967, doi:10.2138/am.2010.3458.

Galuskina, I.O. et al., 2010b, Elbrusite-(Zr)—A new uranian garnet from the Upper Chegem caldera, Kabardino-Balkaria, Northern Caucasus, Russia: American Mineralogist, v. 95, p. 1172–1181, doi:10.2138/am.2010.3507.

Gaspar, M., Knaack, C., Meinert, L.D., and Moretti, R., 2008, REE in skarn systems: A LA-ICP-MS study of garnets from the Crown Jewel gold deposit: Geochimica et Cosmochimica Acta, v. 72, p. 185–205, doi:10.1016/j.gca.2007.09.033.

Gatewood, M.P., Dragovic, B., Stowell, H.H., Baxter, E.F., Hirsch, D.M., and Bloom, R., 2015, Evaluating chemical equilibrium in metamorphic rocks using major element and Sm–Nd isotopic age zoning in garnet, Townshend Dam, Vermont, USA: Chemical Geology, v. 401, p. 151–168, doi:10.1016/j.chemgeo.2015.02.017.

Ghosh, B., and Morishita, T., 2011, ANDRADITE–UVAROVITE SOLID SOLUTION FROM HYDROTHERMALLY ALTERED PODIFORM CHROMITITE, RUTLAND OPHIOLITE, ANDAMAN, INDIA: The Canadian Mineralogist, v. 49, p. 573–580, doi:10.3749/canmin.49.2.573.

Ghosh, B., Morishita, T., Ray, J., Tamura, A., Mizukami, T., Soda, Y., and Ovung, T.N., 2017, A new occurrence of titanian (hydro)andradite from the Nagaland ophiolite, India: Implications for element mobility in hydrothermal environments: Chemical Geology, v. 457, p. 47–60, doi:10.1016/j.chemgeo.2017.03.012.

Gurney, J., and Moore, R.O., 1986, Mineral Inclusions in Diamonds From the Monastery Kimberlite, South Africa, *in* v. 14.

Harangi, 2001, Almandine Garnet in Calc-alkaline Volcanic Rocks of the Northern Pannonian Basin (Eastern–Central Europe): Geochemistry, Petrogenesis and Geodynamic Implications: Journal of Petrology, v. 42, p. 1813–1843, doi:10.1093/petrology/42.10.1813.

Haynes, E.A., Moecher, D.P., and Spicuzza, M.J., 2003, Oxygen isotope composition of carbonates, silicates, and oxides in selected carbonatites: constraints on crystallization temperatures of carbonatite magmas: Chemical Geology, v. 193, p. 43–57, doi:10.1016/S0009-2541(02)00244-9.

Hilton, E., 2000, Composition and structure of titanian andradite from magmatic and hydrothermal environments: University of British Columbia, doi:10.14288/1.0089519.

Hode Vuorinen, J., Hålenius, U., Whitehouse, M.J., Mansfeld, J., and Skelton, A.D.L., 2005, Compositional variations (major and trace elements) of clinopyroxene and Ti-andradite from pyroxenite, ijolite and nepheline syenite, Alnö Island, Sweden: Lithos, v. 81, p. 55–77, doi:10.1016/j.lithos.2004.09.021.

Huang, G., Guo, J., Cui, W., and Palin, R., 2020, Deciphering garnet genesis in meta-igneous rocks: An example from the Jiao-Liao-Ji Belt, North China Craton: Precambrian Research, v. 348, p. 105871, doi:10.1016/j.precamres.2020.105871.

Hutchison, M.T., 1997, Constitution of the deep transition zone and lower mantle shown by diamonds and their inclusions: , p. 856.

Inglis, J.D., Hefferan, K., Samson, S.D., Admou, H., and Saquaque, A., 2017, Determining age of Pan African metamorphism using Sm-Nd garnet-whole rock geochronology and phase equilibria modeling in the Tasriwine ophiolite, Sirwa, Anti-Atlas Morocco: Journal of African Earth Sciences, v. 127, p. 88–98, doi:10.1016/j.jafrearsci.2016.06.021.

Ivanova, O.A., Logvinova, A.M., and Pokhilenko, N.P., 2017, Inclusions in diamonds from Snap Lake kimberlites (Slave Craton, Canada): Geochemical features of crystallization: Doklady Earth Sciences; Dordrecht, v. 474, p. 490–493, doi:http://dx.doi.org.mutex.gmu.edu/10.1134/S1028334X17050129.

Jamtveit, B., Dahlgren, S., and Austrheim, H., 1997, High-grade contact metamorphism of calcareous rocks from the Oslo Rift, Southern Norway: American Mineralogist, v. 82, p. 1241–1254, doi:10.2138/am-1997-11-1219.

Jeong, G.Y., and Kim, Y.H., 1999, Goldmanite from the black slates of the Ogcheon belt, Korea: Mineralogical Magazine, v. 63, p. 253–256, doi:10.1180/002646199548358.

Kaminsky, F., Zakharchenko, O.D., Davies, R., Griffin, W., Khachatryan-Blinova, G.K., and Shiryaev, A., 2001, Superdeep diamonds from the Juina area, Mato Grosso State, Brazil: Contributions to Mineralogy and Petrology, v. 140, p. 734–753, doi:10.1007/s004100000221.

Katerinopoulou, 2009, A multi-analytical study of the crystal structure of unusual Ti–Zr–Cr-rich Andradite from the Maronia skarn, Rhodope massif, western Thrace, Greece: Mineralogy and Petrology, v. 95, p. 113–124, doi:10.1007/s00710-008-0023-4.

Kawakami, T., Horie, K., Hokada, T., Hattori, K., and Hirata, T., 2019, Disequilibrium REE compositions of garnet and zircon in migmatites reflecting different growth timings during single metamorphism (Aoyama area, Ryoke belt, Japan): Lithos, v. 338–339, p. 189–203, doi:10.1016/j.lithos.2019.04.021.

Kebede, T., Koeberl, C., and Koller, F., 2001, Magmatic evolution of the suqii-wagga garnet-bearing two-mica granite, wallagga area, western Ethiopia: Journal of African Earth Sciences, v. 32, p. 193–221, doi:10.1016/S0899-5362(01)90004-1.

von Knorring, O., Condliffe, E., and Tong, Y.L., 1986, Some mineralogical and geochemical aspects of chromium-bearing skarn minerals from northern Karelia, Finland: Bulletin of the Geological Society of Finland, v. 58, p. 277–292, doi:10.17741/bgsf/58.1.019.

Korinevsky, V.G., 2015, Spessartine-Andradite In Scapolite Pegmatite, Ilmeny Mountains, Russia: The Canadian Mineralogist, v. 53, p. 623–632, doi:10.3749/canmin.4354.

Kotková, J., and Harley, S.L., 2010, Anatexis during High-pressure Crustal Metamorphism: Evidence from Garnet–Whole-rock REE Relationships and Zircon–Rutile Ti–Zr Thermometry in Leucogranulites from the Bohemian Massif: Journal of Petrology, v. 51, p. 1967–2001, doi:10.1093/petrology/egq045.

Kozror Solution propertiesof almandine-pyropegarnet as determinedby phaseequilibrium experiments:

Krippner, A., Meinhold, G., Morton, A.C., Schönig, J., and von Eynatten, H., 2016, Heavy minerals and garnet geochemistry of stream sediments and bedrocks from the Almklovdalen area, Western Gneiss Region, SW Norway: Implications for provenance analysis: Sedimentary Geology, v. 336, p. 96–105, doi:10.1016/j.sedgeo.2015.09.009.

Lehnert, K., Su, Y., Langmuir, C., Sarbas, B., and Nohl, U., 2000, A global geochemical database structure for rocks, Geochemistry Geophysics Geosystems, 1, http://dx.doi.org/10.1029/1999GC000026.

Li, D., Fu, Y., and Sun, X., 2018a, Onset and duration of Zn—Pb mineralization in the Talate Pb—Zn (—Fe) skarn deposit, NW China: Constraints from spessartine U—Pb dating: Gondwana Research, v. 63, p. 117–128, doi:10.1016/j.gr.2018.05.013.

Li, X., Song, S., Zhang, L., and Höfer, E.H., 2018b, Application of microprobe-based flank method analysis of Fe3+ in garnet of North Qilian eclogite and its geological implication: Science Bulletin, v. 63, p. 300–305, doi:10.1016/j.scib.2018.01.025.

Locock, A.J., 2008, An Excel spreadsheet to recast analyses of garnet into end-member components, and a synopsis of the crystal chemistry of natural silicate garnets: Computers & Geosciences, v. 34, p. 1769–1780, doi:10.1016/j.cageo.2007.12.013.

Major and trace elements in pyrope–almandine garnets as sediment provenance indicators of the Lower Carboniferous Culm sediments, Drahany Uplands, Bohemian Massif, 2005, Lithos, v. 82, p. 51–70, doi:10.1016/j.lithos.2004.12.006.

Makrygina, V.A., and Suvorova, L.F., 2011, Spessartine in the greenschist facies: Crystallization conditions: Geochemistry International, v. 49, p. 299–308, doi:10.1134/S0016702911030074.

Manton, R.J., Buckman, S., Nutman, A.P., Bennett, V.C., and Belousova, E.A., 2017, U-Pb-Hf-REE-Ti zircon and REE garnet geochemistry of the Cambrian Attunga eclogite, New England Orogen, Australia: Implications for continental growth along eastern Gondwana: Orogens and Oceanic Terranes: Tectonics, v. 36, p. 1580–1613, doi:10.1002/2016TC004408.

Marks, M.A.W., Coulson, I.M., Schilling, J., Jacob, D.E., Schmitt, A.K., and Markl, G., 2008, The effect of titanite and other HFSE-rich mineral (Ti-bearing andradite, zircon, eudialyte) fractionation on the geochemical evolution of silicate melts: Chemical Geology, v. 257, p. 153–172, doi:10.1016/j.chemgeo.2008.09.002.

Mason, B., 1966, Pyrope, augite, and hornblende from Kakanui, New Zealand: New Zealand Journal of Geology and Geophysics, v. 9, p. 474–480, doi:10.1080/00288306.1966.10422491.

McAloon, B.P., and Hofmeister, A.M., 1995, Single-crystal IR spectroscopy of grossular-andradite garnets: American Mineralogist, v. 80, p. 1145–1156, doi:10.2138/am-1995-11-1205.

Mueller, A.G., and Delor, C.P., 1991, Goldmanite-rich garnet in skarn veins, Southern Cross greenstone belt, Yilgarn Block, Western Australia: Mineralogical Magazine, v. 55, p. 617–620, doi:10.1180/minmag.1991.055.381.15.

Munno, R., Rossi, G., and Tadini, C., 1980, Crystal chemistry of kimzeyite from Stromboli, Aeolian Islands, Italy: American Mineralogist, v. 65, p. 188–191.

Naimo, D., Balassone, G., Beran, A., Amalfitano, C., Imperato, M., and Stanzione, D., 2003, Garnets in volcanic breccias of the Phlegraean Fields (southern Italy): mineralogical, geochemical and genetic features: Mineralogy and Petrology; Wien, v. 77, p. 259–270, doi:http://dx.doi.org.mutex.gmu.edu/10.1007/s00710-002-0219-y.

Nixon, P.H., Knorring, O. von, and Rooke, J.M., 1963, Kimberlites And Asociated Inclusions Of Basytoland: A Mineralogical And Geochemical Study: American Mineralogist, v. 48, p. 1090–1132.

Padovani, E.R., and Tracy, R.J., 1981, A pyrope-spinel (alkremite) xenolith from Moses Rock Dike: first known North American occurrence: American Mineralogist, v. 66, p. 741–745.

Pal, T., and Das, D., 2010, Uvarovite from chromite-bearing ultramafic intrusives, Orissa, India, a crystal-chemical characterization using 57Fe Mössbauer spectroscopy: American Mineralogist, v. 95, p. 839–843, doi:10.2138/am.2010.3328.

Parthasarathy, G., Balaram, V., and Srinivasan, R., 1999, Characterization of green garnets from an Archean calc-silicate rock, Bandihalli, Karnataka, India: evidence for a continuous solid solution between uvarovite and grandite: Journal of Asian Earth Sciences, v. 17, p. 345–352, doi:10.1016/S0743-9547(98)00064-6.

Patranabis-Deb, S., Schieber, J., and Basu, A., 2009, Almandine garnet phenocrysts in a ~1 Ga rhyolitic tuff from central India: Geological Magazine, v. 146, p. 133–143, doi:10.1017/S0016756808005293.

Philpotts, J.A., Schnetzler, C.C., and Thomas, H.H., 1972, Petrogenetic implications of some new geochemical data on eclogitic and ultrabasic inclusions: Geochimica et Cosmochimica Acta, v. 36, p. 1131–1166, doi:10.1016/0016-7037(72)90096-8.

Plümper, O., Beinlich, A., Bach, W., Janots, E., and Austrheim, H., 2014, Garnets within geode-like serpentinite veins: Implications for element transport, hydrogen production and life-supporting environment formation: Geochimica et Cosmochimica Acta, v. 141, p. 454–471, doi:10.1016/j.gca.2014.07.002.

Preston, J., Hartley, A., Mange-Rajetzky, M., Hole, M., May, G., Buck, S., and Vaughan, L., 2002, The Provenance of Triassic Continental Sandstones from the Beryl Field, Northern North Sea: Mineralogical, Geochemical, and Sedimentological Constraints: Journal of Sedimentary Research, v. 72, p. 18–29, doi:10.1306/042201720018.

Pribavkin, S.V., Avdonina, I.S., and Zamyatin, D.A., 2013, Mineralogy, conditions of crystallization and melt generation of epidote-bearing porphyries from the Middle Urals, Russian federation: Mineralogy and Petrology, v. 107, p. 125–147, doi:10.1007/s00710-012-0226-6.

Quartieri, S., Boscherini, F., Chaboy, J., Dalconi, M.C., Oberti, R., and Zanetti, A., 2002, Characterization of trace Nd and Ce site preference and coordination in natural melanites: a combined X-ray diffraction and high-energy XAFS study: Physics and Chemistry of Minerals, v. 29, p. 495–502, doi:10.1007/s00269-002-0251-9.

Rahmani Javanmard, S., Tahmasbi, Z., Ding, X., Ahmadi Khalaji, A., and Hetherington, C.J., 2018, Geochemistry of garnet in pegmatites from the Boroujerd Intrusive Complex, Sanandaj-Sirjan Zone, western Iran: implications for the origin of pegmatite melts: Mineralogy and Petrology, v. 112, p. 837–856, doi:10.1007/s00710-018-0591-x.

Reinecke, T., Okrusch, M., and Richter, P., 1985, Geochemistry of ferromanganoan metasediments from the Island of Andros, Cycladic Blueschist Belt, Greece: Chemical Geology, v. 53, p. 249–278, doi:10.1016/0009-2541(85)90074-9.

Russell, J.K., Dipple, G.M., Lang, J.R., and Lueck, B., 1999, Major-element discrimination of titanian andradite from magmatic and hydrothermal environments: an example from the Canadian Cordillera: European Journal of Mineralogy, v. 11, p. 919–936, doi:10.1127/ejm/11/6/0919.

Sajeev, K., Windley, B.F., Connolly, J.A.D., and Kon, Y., 2009, Retrogressed eclogite (20kbar, 1020°C) from the Neoproterozoic Palghat–Cauvery suture zone, southern India: Precambrian Research, v. 171, p. 23–36, doi:10.1016/j.precamres.2009.03.001.

Salnikova, E.B., Chakhmouradian, A.R., Stifeeva, M.V., Reguir, E.P., Kotov, A.B., Gritsenko, Y.D., and Nikiforov, A.V., 2019, Calcic garnets as a geochronological and petrogenetic tool applicable to a wide variety of rocks: Lithos, v. 338–339, p. 141–154, doi:10.1016/j.lithos.2019.03.032.

Schertl, H.-P., Polednia, J., Neuser, R.D., and Willner, A.P., 2018, Natural End Member Samples of Pyrope and Grossular: A Cathodoluminescence-Microscopy and -Spectra Case Study: Journal of Earth Science, v. 29, p. 989–1004, doi:10.1007/s12583-018-0842-0.

Schingaro, 2016, Crystal chemistry and light elements analysis of Ti-rich garnets: American Mineralogist, v. 101, p. 371–384, doi:10.2138/am-2016-5439.

Schingaro, E., Scordari, F., Capitanio, F., Parodi, G., Smith, D.C., and Mottana, A., 2001, Crystal chemistry of kimzeyite from Anguillara, Mts. Sabatini, Italy: European Journal of Mineralogy, v. 13, p. 749–759, doi:10.1127/0935-1221/2001/0013-0749.

Schönig, J., Meinhold, G., von Eynatten, H., and Lünsdorf, N.K., 2018, Provenance information recorded by mineral inclusions in detrital garnet: Sedimentary Geology, v. 376, p. 32–49, doi:10.1016/j.sedgeo.2018.07.009.

Sibi, N., and Subodh, G., 2017, Structural and Microstructural Correlations of Physical Properties in Natural Almandine-Pyrope Solid Solution: Al70Py29: Journal of Electronic Materials, v. 46, p. 6947–6956, doi:10.1007/s11664-017-5801-5.

Sieck, P., López-Doncel, R., Dávila-Harris, P., Aguillón-Robles, A., Wemmer, K., and Maury, R.C., 2019, Almandine garnet-bearing rhyolites associated to bimodal volcanism in the Mesa Central of

Mexico: Geochemical, petrological and geochronological evolution: Journal of South American Earth Sciences, v. 92, p. 310–328, doi:10.1016/j.jsames.2019.03.018.

Sipahi, F., Akpınar, İ., Eker, Ç.S., Kaygusuz, A., Vural, A., and Yılmaz, M., 2017, Formation of the Eğrikar (Gümüşhane) Fe–Cu skarn type mineralization in NE Turkey: U–Pb zircon age, lithogeochemistry, mineral chemistry, fluid inclusion, and O-H-C-S isotopic compositions: Journal of Geochemical Exploration, v. 182, p. 32–52, doi:10.1016/j.gexplo.2017.08.006.

Sipahi, Ferkan, Akpinar, İbrahim, Ekar, Çiğdem Saydam, Kaygusuz, Abdullah, Vural, Alaaddin, and Yılmaz, Meltem, 2017, Formation of the Eğrikar (Gümüşhane) Fe–Cu skarn type mineralization in NE Turkey: U–Pb zircon age, lithogeochemistry, mineral chemistry, fluid inclusion, and O-H-C-S isotopic compositions: ResearchGate, https://www.researchgate.net/publication/318964393_Formation_of_the_Egrikar_Gumushane_Fe-Cu_skarn_type_mineralization_in_NE_Turkey_U-Pb_zircon_age_lithogeochemistry_mineral_chemistry_fluid_inclusion_and_O-H-C-S_isotopic_compositions (accessed March 2019).

Sobolev, N.V., Logvinova, A.M., Zedgenizov, D.A., Seryotkin, Y.V., Yefimova, E.S., Floss, C., and Taylor, L.A., 2004, Mineral inclusions in microdiamonds and macrodiamonds from kimberlites of Yakutia: a comparative study: Lithos, v. 77, p. 225–242, doi:10.1016/j.lithos.2004.04.001.

Sobolev, N.V., Schertl, H.-P., Valley, J.W., Page, F.Z., Kita, N.T., Spicuzza, M.J., Neuser, R.D., and Logvinova, A.M., 2011, Oxygen isotope variations of garnets and clinopyroxenes in a layered diamondiferous calcsilicate rock from Kokchetav Massif, Kazakhstan: a window into the geochemical nature of deeply subducted UHPM rocks: Contributions to Mineralogy and Petrology, v. 162, p. 1079, doi:10.1007/s00410-011-0641-4.

Song, S., Yang, J., Liou, J.G., Wu, C., Shi, R., and Xu, Z., 2003, Petrology, geochemistry and isotopic ages of eclogites from the Dulan UHPM Terrane, the North Qaidam, NW China: Lithos, v. 70, p. 195–211, doi:10.1016/S0024-4937(03)00099-9.

Stachel, T., Harris, J., and Brey, G., 1998, Rare and unusual mineral inclusions in diamonds from Mwadui, Tanzania: Contributions to Mineralogy and Petrology, v. 132, p. 34–47, doi:10.1007/s004100050403.

Stachel, T., Brey, G.P., and Harris, J.W., 2000, Kankan diamonds (Guinea) I: from the lithosphere down to the transition zone: ResearchGate, doi:http://dx.doi.org/10.1007/s004100000173.

Stähle, V., Altherr, R., Nasdala, L., and Ludwig, T., 2011, Ca-rich majorite derived from high-temperature melt and thermally stressed hornblende in shock veins of crustal rocks from the Ries impact crater (Germany): Contributions to Mineralogy and Petrology; Heidelberg, v. 161, p. 275–291, doi:http://dx.doi.org.mutex.gmu.edu/10.1007/s00410-010-0531-1.

Stalder, M., and Rozendaal, A., 2005, CALDERITE-RICH GARNET AND FRANKLINITE-RICH SPINEL IN AMPHIBOLITE-FACIES HYDROTHERMAL SEDIMENTS, GAMSBERG Zn-Pb DEPOSIT, NAMAQUA PROVINCE, SOUTH AFRICA: The Canadian Mineralogist, v. 43, p. 585–599, doi:10.2113/gscanmin.43.2.585.

Suwa, K., Suzuki, K., and Agata, T., 1996, Vanadium grossular from the Mozambique metamorphic rocks, south Kenya: Journal of Southeast Asian Earth Sciences, v. 14, p. 299–308, doi:10.1016/S0743-9547(96)00066-9.

Taran, M.N., and Langer, K., 2000, Electronic absorption spectra of Fe3+ in andradite and epidote at different temperatures and pressures: European Journal of Mineralogy, v. 12, p. 7–15, doi:10.1127/0935-1221/2000/0012-0007.

Thomson, A.R., Kohn, S.C., Bulanova, G.P., Smith, C.B., Araujo, D., and Walter, M.J., 2016, Trace element composition of silicate inclusions in sub-lithospheric diamonds from the Juina-5 kimberlite: Evidence for diamond growth from slab melts: Lithos, v. 265, p. 108–124, doi:10.1016/j.lithos.2016.08.035.

Tian, Z.-D., Leng, C.-B., Zhang, X.-C., Zafar, T., Zhang, L.-J., Hong, W., and Lai, C.-K., 2019, Chemical composition, genesis and exploration implication of garnet from the Hongshan Cu-Mo skarn deposit, SW China: Ore Geology Reviews, v. 112, p. 103016, doi:10.1016/j.oregeorev.2019.103016.

Volkova, N.I., Kovyazin, S.V., Stupakov, S.I., Simonov, V.A., and Sakiev, K.S., 2014, Trace element distribution in mineral inclusions in zoned garnets from eclogites of the Atbashi Range (South Tianshan): Geochemistry International, v. 52, p. 939–961, doi:10.1134/S0016702914090092.

Walter, M.J., Nakamura, E., Trønnes, R.G., and Frost, D.J., 2004, Experimental constraints on crystallization differentiation in a deep magma ocean: Geochimica et Cosmochimica Acta, v. 68, p. 4267–4284, doi:10.1016/j.gca.2004.03.014.

Wang, L., Essene, E.J., and Zhang, Y., 2000, Direct observation of immiscibility in pyrope-almandine-grossular garnet: American Mineralogist, v. 85, p. 41–46, doi:10.2138/am-2000-0106.

Wang, L., Essene, E.J., and Zhang, Y., 1999, Mineral inclusions in pyrope crystals from Garnet Ridge, Arizona, USA: implications for processes in the upper mantle: Contributions to Mineralogy and Petrology, v. 135, p. 164–178, doi:10.1007/s004100050504.

Weiss, J., 1949, WISSAHICKON SCHIST AT PHILADELPHIA, PENNSYLVANIA: GSA Bulletin, v. 60, p. 1689–1726, doi:10.1130/0016-7606(1949)60[1689:WSAPP]2.0.CO;2.

Wilding, M.C., 1990, A study of diamonds with syngenetic inclusions.

Yang, J., Peng, J., Hu, R., Bi, X., Zhao, J., Fu, Y., and Shen, N.-P., 2013, Garnet geochemistry of tungsten-mineralized Xihuashan granites in South China: Lithos, v. 177, p. 79–90, doi:10.1016/j.lithos.2013.06.008.

Zedgenizov, D.A., Kagi, H., Shatsky, V.S., and Ragozin, A.L., 2014, Local variations of carbon isotope composition in diamonds from São-Luis (Brazil): Evidence for heterogenous carbon reservoir in sublithospheric mantle: Chemical Geology, v. 363, p. 114–124, doi:10.1016/j.chemgeo.2013.10.033.

Zeh, A., and Gerdes, A., 2014, HFSE (High Field Strength Elements)-transport and U–Pb–Hf isotope homogenization mediated by Ca-bearing aqueous fluids at 2.04Ga: Constraints from zircon, monazite, and garnet of the Venetia Klippe, Limpopo Belt, South Africa: Geochimica et Cosmochimica Acta, v. 138, p. 81–100, doi:10.1016/j.gca.2014.04.015.

Zhang, X.-Y., Wang, H., and Yan, Q.-H., 2022, Garnet geochemical compositions of the Bailongshan lithium polymetallic deposit in Xinjiang Province: Implications for magmatic-hydrothermal evolution: Ore Geology Reviews, v. 150, p. 105178, doi:10.1016/j.oregeorev.2022.105178.

Zhou, J., Feng, C., and Li, D., 2017, Geochemistry of the garnets in the Baiganhu W–Sn orefield, NW China: Ore Geology Reviews, v. 82, p. 70–92, doi:10.1016/j.oregeorev.2016.11.019.

---

## Author Response (AR1)

**Garnet Manuscript Reviewer Comments**

All responses to the reviewers from the authors are indicated in **bold text**.

**Reviewer 1:**

**Citation**: https://doi.org/10.5194/essd-2023-45-RC1

General Comments

The manuscript reports on a database of chemical analyses, ages, localities, paragenesis, P-T conditions etc. of garnets and some preliminary interpretations.

1) The dataset could be indeed be very interesting if obviously wrong entries would be eliminated, and a more convincing quality control would be used, e.g. by calculation of garnet species or end-members from the analyses.
Only a very small part of the data presented in the database is new and original. The quality of the new data is very good. However, the authors have also included analyses of inclusion minerals as "garnet" analyses in the database. I cannot see any use of including such obvious wrong analyses in the data set.

A similar problem is that the authors include data from older peer reviewed publications (the "dark data") that are clearly in the original publication not identified as garnet, but as whole rock analyses of garnet-bearing rocks, such as quartzites. These include:
Project ID 43-52 are not spessartine mineral analyses but so-called "coticules", i.e. garnet-bearing quartzites and one of the authors, KC, has even included bulk analyses of slates and volcanic rocks as "garnet" analyses (see Herbosch et al. 2016, Table 3). Project ID 146-171 are quartzite whole rock analyses, not garnets (Reinecke et al. 1986 not Reincke as stated in the database).
Thus, whole rock analyses were included in the data set as mineral analyses. These were avoidable errors of the authors. These inconsistent data must be eliminated.

**We agree with the reviewer that these are erroneous samples and should not be included. We did a thorough check of the dataset and included all changes below:**

**Notes:**
- **Deleted samples from Herbosch et al 2016 (Project ID 43-52).**
- **Deleted sampled from Reinecke et al 1986 (Project Id 146 - 171).**
- **Added more specific paragenesis to Schönig et al. (2018) (Project ID 1081 - 1373).**
- **Added 4 Grossular samples from Naimo et al (2003).**
- **Andradite samples in Katerinopoulou (2009) were excluded in silica confidence interval.**
- **Andradite samples had a difficult time with the silica confidence interval (range of SiO2: 29 - 32 wt%).**
- **Added temperature data to Plümper et al. (2014).**

- Russell et al. (1999) has more data but cannot find repository: A database of titanian andradite compositions is available from JKR or GMD or via ftp [http://perseus.geology.ubc.ca/].
- Updated the Russell et al. (1999) Geological Context column (Project ID 1587-1590).
- Added 3 rim analyses from Marks et al. (2008) (Project ID's: 1620, 1623, 1625).
- Updated the following columns for both Marks et al. 2008 papers (Project ID 1606-1625): Notes, Mineral, Zone, Area, Geological Context, Analysis Method.
- Updated the Title and Journal columns for Marks et al. (2008) (Project ID 1606-1617).
- Silica confidence interval excluded kimzeyites from Beard and Drake (2007) (range of SiO2: 23 - 35 wt%).
- Sieck et al. (2019) paragenesis of "rhyolitic flows, ignimbrite" -> "Rhyolite" and added P&T data.
- Pribavkin, Avdonina, and Zamyatin (2012) added samples (origin ID) 16, 17, & 18.
- Added 20 samples to Patranabis-Deb, Schieber, and Basu (2009).
- Deleted repeat of Beard & Drake (2007) (Project ID 1987 - 1995; 58-66 MG).
- Deleted repeat of Jamtveit et al. (1997) and Naimo et al. (2003).
- Von Knorring (1986) changed paragenesis from "Unknown" to "Skarn".
- Mueller and Delor (1991) changed paragenesis from "Skarn veins" to "Skarn".
- Ghosh and Morishita (2011) hydrothermal alteration of the peridotite led to garnet formation -> updated paragenesis.
- Added Age, Temp, Pressure to Kotkova and Harley (2010)
- Updated paragenesis in Inglis et al. (2017) and Zeh & Gerdes (2014)..
- Added age, temp, pressure and paragenesis to Wang et al. (1999).
- Added temp and pressure to Enami et al. (1993).
- Added garnet species names to literature from Locock (2008) but there is still no age, temp, pressure, or paragenesis.
- Marked Project ID 3314-3330 and 3461-3463 as repeats (they are part of Locock (2008)'s dataset.
- Updated varietal name for Project ID: 2085-2095.
- The Kimzeyites and Elbrusites in Galuskina et al (2010) are excluded from SiO2 intervals (anomalously low SiO2 ~3 wt. % for high UO3 ~20 wt. %).
- Added age, temp, pressure to Kawakami et al. (2019).
- Added 17 foliated eclogite samples to Li et al. (2018).
- Added Age, Paragenesis, and Garnet name to Salnikova et al. (2019), SiO2 excludes some of these andradites.
- Added temperature to Salnikova et al. (2019).
- Added 95 samples from Philpotts et al. (1972), Huang et al. (2020), Zhang et al. (2022).
- Deleted original EMPA inclusions: chromite, dark almandine, light almandine, dark uvarovite.
- Fixed total and our total wt. % columns.
- Removed the Silica Confidence Interval.
- Evaluated all the garnet sample analyses and reclassified them using the spreadsheets from Locock (2008) and Grew et al. (2013) to calculate the end-member species and provide a Quality Index from the geochemical data.

2) The authors use a "Silica Confidence Interval" (SCI) method to exclude samples of questionable composition from further analysis. This method seems to identify the above mentioned whole rock analyses of quartzites as unlikely of garnets and analyses of minerals that are not garnets but pyroxenes or spinel group minerals from the EarthChem database. However, it also seems to eliminate analyses, e.g. of henritermierites (Project ID 60-61), titanian andradites, schorlomites, kimzeyites, katoite-rich (hydro)grossulars of high quality! Thus, the SCI method is not very useful if the mineral species is not considered. Partial analysis of inclusions and garnet, a concern of the authors, will often not be identified correctly by this method.

A much better method to evaluate the quality of garnet analyses would be to calculate the end-member species from the chemical analysis using the approach of Locock (2008) and Grew et al. (2013) and calculate a "Quality Index" as suggested by Locock (2008). See Hawthorne (2021, Can. Mineral. 59, 169ff).

**We agree with this comment from the reviewer and evaluated all of the garnets using a combination of the spreadsheets from Locock (2008) and Grew et al. (2013). In the list above, we flagged some examples of samples that were erroneously excluded using the silica confidence interval. We decided to instead, remove erroneous samples (such as Herbosch et al. (2016) and Reinecke et al. (1986)) with the full list of dataset changes listed above. We eliminated the silica confidence interval since we should not exclude the samples of low SiO2 wt% from the dataset such as kimzeyites and elbrusites (Galuskina et al., 2010).**

3) The discussion and interpretation of the data set is focused mainly on frequency plots of major (and some minor) elements and on the binary correlations of elements for various "material types" (igneous, metamorphic, detrital and unknown), in my view, the least reliable categorization of garnets (see below).

The dataset is heavily biased by garnets from the mantle (mostly brought to the surface by volcanic rocks) and by a study of garnets in a single amphibolite from the crust. The authors do not eliminate these overrepresented data in their data evaluation or weight them accordingly. Thus, any meaningful evaluation of the data must consider or correct for the bias.

The interpretation of the data, the main part of the manuscript, is therefore not very insightful. The interpretation of observed correlations between two elements as binary series of two garnet species on the other hand, is trivial if only binary correlations are studied. Why not use multivariate statistical methods or explore ternary compositions? The lengthy discussion of the binary element correlations and the frequency plots of strongly biased data stands in contrast to the few new insights gained from the analysis of the database. Some of the conclusions are probably wrong (see below).

**We thank the reviewer for the thoughtful suggestion. To clarify, the purpose of this paper is solely to be a data description paper, rather than a multivariate analysis of the data present. The correlation coefficient plots were intended to solely show any users of the dataset what data and geochemical samples are currently present in this version of the dataset. Given that the purpose of these plots caused confusion for the reviewer, we have removed them and any discussion relating to them from the paper. We do not intend to analyze or draw any conclusions from the data. We intend only to provide this dataset to future researchers who may wish to use it for their own work or to upload their own garnet geochemical analyses to a larger mineralogical repository.**

**Further, we did initially create ternary diagrams of the data to show the distributions of garnet geochemistry present. However, our coauthor Frank Spear found significant biases by using ternary diagrams based on the idealized end-member species and**

**proposed a more thorough way to visualize the distributions is by creating correlation coefficient plots of the measured major oxides. Therefore, any researcher who needed to identify a range of geochemical properties would have a better way to interpret whether this dataset is useful to them. We have removed this discussion based on the reviewer's feedback.**

4) The discrimination between an igneous and a metamorphic origin for mantle garnets is a question of semantics and ambiguous. I would rather suggest that the authors discuss garnets from distinct "paragenesis" instead of their "material types". See for example the approach of Krippner et al. (2014 Sed. Geol. 306, 36ff) - a relevant publication not cited by the authors. Igneous versus metamorphic origin of Earth's mantle materials: The authors (and the sources they use) classify all ultramafic/peridotitic materials as "igneous" and all eclogites as "metamorphic". This is an arbitrary decision. If a fertile mantle lherzolite is partially molten, a basaltic liquid extracted and then crystallizes within the upper mantle, it will have an eclogite-like mineral paragenesis consisting of pyrope- and grossular-rich almandine garnet and an omphacitic clinopyroxene. The authors will classify the rock and garnet as "metamorphic", although it is obviously an igneous rock and mineral. An ultramafic rock of bridgmanite-ferropericlase composition from the lower mantle that is brought by diapirism into the upper mantle will be transformed in solid state thus by a metamorphic process into a rock of garnet-bearing peridotite paragenesis. Thus, the discrimination between (and discussion of) compositions of igneous and metamorphic rocks makes little sense in the realm of mantle rocks, the overwhelming lithology in the database. This problem can also be seen in the material type classification of majorite analyses from inclusions in diamonds (see below). Thus, similar to the authors' use of the class "detrital", I would suggest that the authors use the term "mantle" in the category "Material". But it is recommended that the authors discuss the garnet compositions of distinct paragenetic assemblages not the ambiguous "Materials".

**We agree with the reviewer that this classification distinction is a little complicated, however, we adopted this column "Material" directly from the EarthChem repository. For the sake of data continuity, we would like to respect their decisions and data classification of their 61,294 samples and maintain this column and all designations. We agree with the reviewer that the classification of igneous vs metamorphic in the example they describe is highly subjective and complicates the interpretation of this data. We recommend that users of the dataset keep this limitation in mind, and we highlighted this limitation in the methods section of the text: "We recommend examining each of the petrogenetic attributes collectively as well as individually to best characterize the data with cluster analysis. It should also be noted that how each of the attributes are classified remains a subject of debate as they are highly subjective and vary over time and between authors. For example, the distinction between igneous and metamorphic rocks can be arbitrary when various mantle processes at various depths can be responsible for a specific rock's minerology and texture."**

**Similarly, they classified all of their samples as a general "garnet" for the mineral name, therefore we must maintain this classification and instead added an additional column, "Species," to reflect the mineral species classification based on the spreadsheets from Locock (2008) and Grew et al. (2013) and added a "Quality Index" column. Hopefully these addition addresses some of the reviewer's concerns.**

Specific comments:

line 112: Goldmannite is defined as the Ca3V3+2Si3O12 endmember, not as Ca3[V,Al,Fe,Ti]2Si3O12. It might (and it always does) contain additionally tri- or tetravalent cations in the octahedral site such as Al, Fe3+ and Ti4+ or very rarely Ti3+, but these elements are not essential (e.g. Grew et al. 2013) and should not be reported in the formula of a mineral species.

**Accepted and incorporated: "There are also reported rare instances of goldmanite ($Ca_3V^{3+}_2Si_3O_{12}$), eringaite ($Ca_3Sc_2Si_3O_{12}$), and rubinite ($Ca_3Ti_2Si_3O_{12}$) occurring in chondrite meteorites (Hazen et al., 2008; Grew et al., 2013; Morrison and Hazen, 2020)."**

line 119: First formation of almandine around 4.0 to 3.5 Ga: Some of the Hadean zircons (> 4.0 Ga) are probably derived from felsic continental crust (e.g. Zhong et al. 2023 Comm. Earth and Env. https://doi.org/10.1038/s43247-023-00731-7 and references therein) that could also contain almandine. I can see no indication for this late suggested appearance of almandine (and spessartine).

**Accepted and incorporated: "Almandine ($Fe_3Al_2Si_3O_{12}$) possibly first formed around 4.4 to 3.3 Ga as it is indicative of felsic igneous environments, occurs in medium- to low-grade metamorphic terrains and is typically found in pegmatites, granite, mica schist, or gneiss (Deer et al., 1982; Nesse, 2013; Zhong et al., 2023). A transition from stagnant lid to present day active lid plate tectonics occurred between 4.4-2.5 Ga (Cawood et al., 2022). The appearance of spessartine ($Mn_3Al_2Si_3O_{12}$), which occurs in uplifted regional metamorphic environments, most likely occurred around 3.6-2.5 Ga during which lateral tectonics initiated and the lithosphere went from variable to uniformly rigid (Hazen et al., 2008; Bauer et al., 2020; Hawkesworth et al., 2020; Cawood et al., 2022)."**

line 124: Uvarovite does not occur or form in "igneous environments". It is rather a typical metasomatic or better hydrothermal mineral (see e.g. Melcher et al. J. Petrol. 38, 1419ff and Farré-de-Pablo et al. 2021 Mineralium Deposita 57, 955ff and references therein).

**Accepted and incorporated: "Uvarovite is rare and occurs in chromite-rich metasomatic or hydrothermal environments (Deer et al., 1982; Farré-de-Pablo et al., 2022; Melcher et al., 1997; Nesse, 2013)."**

line 435: The authors use the category 'almandine-pyrope' for garnets near 50-50 compositions. An approach not supported by the IMA convention, but in my (and some other's) view quite useful in practice. But what is the meaning of - and the reason to include - the category 'pyrope-almandine' then (see e.g. Fig.1)? I would suggest to merge these two categories and those of the other intermediate species with "flipped" composite names.

**We decided to take the reviewers feedback and merge these categories in the dataset for simplicity. We would prefer to list the category that is slightly more prevalent first (i.e., 'pyrope-almandine'), however, we recognize that this complicates the categories and combined them under one notation ('almandine-pyrope').**

**See new text: "There are 37 IMA-recognized structural garnet species and 14 silicate garnets, however, there are 32 categories of mineral names within the dataset which includes the combination of end-members such as 'Almandine-Grossular' and 'Almandine-Pyrope' for samples near 50-50 in composition as well as the simplified term 'Garnet' for unidentified samples. For samples that reported a near 50-50 composition, we standardized the naming convention to one category. For example, sample analyses that reported 'Pyrope-Almandine' are included in 'Almandine-Pyrope' for simplicity."**

Fig.1: The authors should eliminate the following "Mineral" categories, as they are meaningless:

"Andradite-Grandit" (single entry): "grandite" is not a garnet species but an acronym derived from grossular-andradite for garnets of the grossular-andradite join. Thus, andradite-grossular-andradite makes no sense.

"Piemontite-Spessartin" (15 entries): "piemontite is a Mn-rich species of the epidote family not a garnet. The reported analyses have 0.4 to 2.2 % $K_2O$ and more than 70% $SiO_2$ and only very minor MnO and $Mn_2O_3$ concentrations (<3 wt%). This is a very clear misidentification by Chiama et al. The original publication (Reinecke et al. 1985 not Reincke et al. 1985) unambiguously says "piemontite-spessartine and spessartine quartzite". Thus these analyses are bulk XRF analyses of various quartzites and not garnet mineral analyses! The garnet analyses are presented for this locality in Reinecke (1986) in the same journal, but they were not included in the database. Why and how the authors have selected this source?

- **These values were removed.**

line 679-683 and Fig 6a: The discussion of age distributions in terms of mineral evolution is in my view misleading. The overwhelming majority of garnet "ages" relate to mantle xenoliths transported by explosive volcanism to the surface and the reported ages are overwhelmingly the ages of the kimberlite eruptions. As the timing of kimberlite volcanism is in almost all cases unrelated to garnet growth in the mantle rocks, the discussion of age distribution is meaningless for age distribution of garnet growths. Thus, only age values of directly dated garnets should be evaluated here.

**We agree that the garnet "ages" is misleading and few literature sources directly dated the garnet samples and instead provided only general ages of the formation or host rock in the literature. We removed this discussion and figures based on the reviewer's comments that this is "meaningless."**

line 765: For metasomatic garnets, dominated by skarn assemblages, a significant correlation between $Fe^{3+}$ and $Al^{3+}$ is found that is later interpreted as representing the binary substitution between andradite and grossular. The other significant correlation between $Fe^{3+}$ and Si is not discussed. Why? This correlation is simply a consequence of interpreting mass-percentages instead of molar units or endmembers. Andradite with a full occupancy of Si on the tetrahedral site has only 35 mass-% $SiO_2$, while grossular lacking any katoite (or hydrous) component has 40 mass-% $SiO_2$. Thus, the negative correlation of $Fe^{3+}$ and Si is simply a consequence of the interpretation of mass percentages and is not related to any substitution of Si by $Fe^{3+}$.

- **This discussion of the correlation coefficients plots was removed.**

line 927: the moderate to weak correlation for Mg-Si in the "metamorphic matrix" "may be caused by majorite analyses". The correlation coefficient is only 0.126. The 156 majorite analyses in the database are classified as extraterrestrial (4 entries), igneous (28 entries), metamorphic (39 entries) and unknown (85 entries). All igneous and unknown majorites are inclusions in diamonds. 22 of the "metamorphic" majorites are also inclusions in diamonds and the remaining 17 analysis formed in an amphibolite with an impact setting. Here you can see the significant problem of categorizing of materials from the mantle. In the category "metamorphic" 24,601 garnet analyses are plotted and the 39 majorite analyses from a metamorphic setting should be responsible for the moderate to weak correlation between Mg and Si? I doubt that. The correlation is again related to effects of discussing mass-percentages instead of molar proportions. Thus, I strongly recommend to discuss molar proportions or endmembers.

- **This discussion of the correlation coefficients plots was removed.**

line 935-6. "Unknown" matrix: "The Mg - Si relationship represents majorite garnets". Here the correlation coefficient Mg-Si is much higher (0.491). Again, only 85 majorite analysis should influence the correlation of 9476 garnets? It is more probable that garnets with high pyrope

content, the most common Mg-rich endmember garnet, also have higher Si values, as pyrope is the garnet with the highest Si content on a mass or weight basis.
- **This discussion of the correlation coefficients plots was removed.**

References: Where are the full references of all the publications cited in the database? I have checked some and found many typos, especially in the new additions from the authors.
- **A list of references for the dataset itself is added as a supplemental file and these references are also cited directly in the dataset. Corrections and typos were included in version 2 of the dataset and a documentation of the changes is included above.**

**Reviewer 2:**

**All responses to the reviewer from the authors are indicated in bold text. We would like to kindly thank the reviewer for their thorough review of both the dataset as well as the manuscript. Their comments made both significantly stronger and we hope that our revisions are satisfactory.**

**Citation**: https://doi.org/10.5194/essd-2023-45-RC2

This is to present a comprehensive dataset of geochemical characteristics of garnet. The authors have treated 95,588 (!) garnet samples from various sources and compiled their geochemical characters combined with other properties. It is useful without doubt for people who are interested in garnet if used carefully at their responsibility. There are, however, some questions and issues that should be more clearly stated in the dataset.

**We appreciate this reviewer's recognition of the value of our compiled database.**

1. The area of the data sources of garnet analyses. I would greatly appreciate the authors' effort to collect so many garnet analyses, but the area of their search is not so clear. Have they searched garnet analyses in the sources written in English? I have seen many garnet analyses in papers written in non-English languages. What was the authors' strategy for data collection?

**We searched for English-written literature sources only and added a mention of this to the paper. Certainly, there are thousands of more garnet sample analyses that are published in non-English written literature and we certainly would appreciate if non-English speakers donate their data to this repository in the future!**

2. The most serious point is that the attributes the authors used are various in character. Some of them, especially those related to petrogenesis or origin, are highly interpretative and totally depend on authors' interpretation of source literature. They can be, however, changed with time in future studies. In addition, the origins of garnet, igneous, metasomatic and metamorphic, are sometimes difficult to determine, especially in deep-seated rocks.

3. Similar issue. The grouping of "Petrogenesis attributes" (Table 3) are somewhat confusing.  "Type" is composed of five attributes, but four of those (Xenolith, Amphibolite, Xenocryst and Volcanic) are not the same in kind to each other: xenoliths

and xenocrystals are fragments of rocks and minerals, respectively, in igneous (especially volcanic) rocks. So, xenocrysts (and possibly xenoliths) are in part "volcanic". Amphibolite is a name of one of metamorphic facies, and is the name of the rocks of amphibolite facies. Of course, we can find "Xenolith" of "Amphibolite" in "Volcanic" rocks. It is unbelievable for me the largest number of garnet analyses have been from rocks of "Unknown" "Type". This may mean that selection of attributes in terms of "Type" is not appropriate.

**The 'type' of material consists of 56 unique categories, the ones listed by the reviewer are the 5 most prevalent in the dataset listed in table 3 (now table 2). We changed some of the wording and presentation of the dataset to make it clear that there are more options that the 5 listed.**

**Overall, we agree that this is an issue and that the interpretation of the Material, Type, Composition, and Paragenesis attributes are highly subjective and often can vary due to individual interpretation of the literature. We settled on using these categories due to the format of the EarthChem dataset and since all their samples already came with the mentioned classifications and distinctions applied, we did not attempt to reclassify any sample analyses that originate from data repositories for the sake of data continuity. Hopefully authors that use this data resource will carefully consider these limitations and can provide more insight in the future from their geochemical signatures.**

4. The list of attributes as "Paragenesis" in Table 3 is especially embarrassing. "Schist" may represent metamorphic rocks with schistosity irrespective of composition. Others (Kimberlite, Peridotite, Lherzolite and Harzburgite) are classified based on composition or modal assemblage of minerals. There are "schistose peridotites". Is "Peridotite" representative of coarse-grained olivine-rich rocks, which are not referred to mineral assemblage or modal compositions by original authors? It is incredible, possibly meaning some unreliable descriptions. "Lherzolite" and "Harzburgite" are of course included in "Peridotite".

**Again, these are classifications that originated from the EarthChem repository that we did not attempt to alter due to data continuity. We recommend that researchers that use this dataset in the future keep this in mind as it is an important limitation.**

5. I propose to consider "Massif" or something like that instead of "Amphibole" and others to represent non-xenolithic metamorphic or igneous rocks. "Amphibilite" should be included in "Paragenesis" and included in "Schist". "Lherzolite" and "Harzburgite" should be discarded and included in "Peridotite" to avoid confusion.

**We recommend that these distinctions are adopted in future uses of the data, however, EarthChem requested that we maintain data continuity and therefore, we cannot change these classifications in the overarching dataset.**

**We added additional text in the results and discussion section to address the limitations mentioned by the reviewer:**

**"Nevertheless, there are some limitations regarding the classifications of the petrogenetic and paragenetic attributes that must be considered when evaluating this dataset. First, these distinctions are simplified and could be subjective to each authors**

interpretation. For example, within the 'Type' category of 'Xenoliths', these rock fragments could consist of different formation processes (such as fragments of amphibolite/granulite/eclogite facies) that were captured in a volcanic sequence. Thus, their Type as a Xenolith would not represent the individual formation processes of the garnets within the host rock. Second, some classifications of paragenesis do not contain compositional information. For example, a 'Schist' does not consider the compositional origin of the parent rock and therefore could be a peridotite with a foliated texture. Finally, these classifications and distinctions were adopted from the EarthChem repository to maintain data continuity. Therefore, this dataset provides the original classifications applied to the data donated to the repository – presumably from the original authors themselves, although this cannot be guaranteed. For example, while Peridotite is listed as a category within paragenesis, so are Lherzolite and Harzburgite which are types of peridotites. We recommend that these categories be grouped together when analysing this dataset further. Ideally, a system of properly representing the rock-type origin and individual mineral formation processes should be developed to prevent misinterpretation of samples within large datasets such as this one.

There could be other limitations other than the specific examples mentioned here. We recommend that any researchers using this dataset for their own work carefully consider the petrogenetic and paragenetic categories as well as original sources of the data."

---

## Author Response (AR2)

**Topic editor decision: Publish subject to minor revisions (review by editor)**
by Attila Demény
**Public justification (visible to the public if the article is accepted and published)**:
The paper shows a database of garnet compositions, which can not only be used in geological research, but also in archaeological provenance studies.

Additional private note (visible to authors and reviewers only):
Please read the opinion of the reviewer (pasted below) and insert some explanatory sentences in the text as suggested (see the last sentences). Then the paper will be accepted.
* * *
The manuscript focuses on the creation and use of a freely accessible dataset of published analyses of 95,650 different genetic types of garnets from different rocks around the world. Of particular interest are the data published before 1990, which are only available in printed journals. These data usually contain garnets with unusual compositions, endmembers, or garnets with high concentrations of one or more elements. In most cases the localities where the garnet comes from are no more available. The proposed data files can help users for preliminary categorizations and correlation of newly described garnet with those already described in the literature. However, when studying and correlating common garnets in igneous and metamorphic rocks, it is necessary to clarify first what we expect from their comparison. In metamorphic rocks, garnet forms by gradual fractionation of the host rock composition during temperature and pressure increase that results in compositional zoning from the cores to the rims of garnet. This means that analyses with different compositions can come from a single grain, not to mention garnets grains from different compositional layers within the same sample. This also applies to trace elements, some of which have a high partition coefficient in garnet and results in concentration in the cores compared to the rims.

Although garnet is one of the best refractory phases, which can preserve compositional growth zoning under low to medium temperature conditions, by high temperatures imprint its zonation can be modified and homogenized. If the rock was subject to a new metamorphic process, the garnet can undergo decomposition, dissolution, and regrowth. Therefore, it is important to know from which part of the garnet or from which generation the selected analysis comes from. There are number of plutonic rocks that crystallized in the lower-medium crustal levels, where garnet formed during cooling as a result from reactions among the already crystallized minerals and residual melt. In contrast to volcanic rocks, such garnet must not belong to stable phase during overall magma crystallization and cannot be compared with garnet crystallized under equilibrium conditions with other phases.

In summary, it can be said that the proposed data set of garnet analyses is important, especially for garnets with specific (unusual) composition and genetic types, but mainly,

that the set includes garnet analyses from older literature. However, for overall description and explanation in the manuscript, it is necessary to outline the limitations of the data set, especially the fact that the different composition may come from individual garnet grains due to its growth zonation. Further complexity can occur when the zoning profile is modified or garnet underwent partial decomposition, dissolution, and accretion of a new garnet during subsequent metamorphic processes. Such examples are reported throughout the literature and were most recently summarized in (Journal of Petrology, 2023).

**Author Response: Thank you so much for the second review of this paper. We greatly appreciate the time and care provided in these comments and added a section (3.5 Dataset Applications and Limitations) to address these concerns. We also added a citation to the work by Wang et al., 2023 in Journal of Petrology and we hope this is the correct paper as indicated by the reviewer.**